# INSTAREVIVE: ONE-STEP IMAGE ENHANCEMENT VIA DYNAMIC SCORE MATCHING

**Yixuan Zhu**[1]*, **Haolin Wang**[1]*, **Ao Li**[2], **Wenliang Zhao**[1], **Jingxuan Niu**[2],
**Yansong Tang**[2], **Lei Chen**[1,†], **Jie Zhou**[1], **Jiwen Lu**[1]

[1]Department of Automation, Tsinghua University
[2]Tsinghua Shenzhen International Graduate School, Tsinghua University

## ABSTRACT

Image enhancement finds wide-ranging applications in real-world scenarios due to complex environments and the inherent limitations of imaging devices. Recent diffusion-based methods yield promising outcomes but necessitate prolonged and computationally intensive iterative sampling. In response, we propose InstaRevive, a straightforward yet powerful image enhancement framework that employs score-based diffusion distillation to harness potent generative capability and minimize the sampling steps. To fully exploit the potential of the pre-trained diffusion model, we devise a practical and effective diffusion distillation pipeline using dynamic control to address inaccuracies in updating direction during score matching. Our control strategy enables a dynamic diffusing scope, facilitating precise learning of denoising trajectories within the diffusion model and ensuring accurate distribution matching gradients during training. Additionally, to enrich guidance for the generative power, we incorporate textual prompts via image captioning as auxiliary conditions, fostering further exploration of the diffusion model. Extensive experiments substantiate the efficacy of our framework across a diverse array of challenging tasks and datasets, unveiling the compelling efficacy and efficiency of InstaRevive in delivering high-quality and visually appealing results. Code is available at `https://github.com/EternalEvan/InstaRevive`.

## 1 INTRODUCTION

Image enhancement seeks to improve the visual quality since images captured in wild scenarios always suffer from various degradations, like noise, blur, downsampling and compression due to the limitations of current imaging devices and complex environments. In recent years, great development has been achieved in the image enhancement field using deep learning methods (Dong et al., 2014; Zhang et al., 2017; Liang et al., 2021; Chen et al., 2021; Wang et al., 2022b; Zamir et al., 2022; Chen et al., 2023b). While these methods yield commendable outcomes under specific, well-defined degradations, they often fall short when faced with the complex conditions of real-world scenarios. Thus, our basic goal is to construct an effective and robust image enhancement framework capable of addressing various degradation conditions. This framework aims to deliver high-quality, visually appealing results within a limited computational budget, making it more practical for real-world use.

Image enhancement is ill-posed due to unknown degradation processes, allowing for various possible high-quality (HQ) results from low-quality (LQ) inputs. To address this problem, researchers have explored a wide range of deep learning methods, which can be generally categorized into three classes: predictive (Huang et al., 2020; Gu et al., 2019; Zhang et al., 2018a), GAN-based (Wang et al., 2021a; Yuan et al., 2018; Fritsche et al., 2019; Zhang et al., 2021; Wang et al., 2021c) and diffusion-based (Kawar et al., 2022; Wang et al., 2022a; Fei et al., 2023; Lin et al., 2023b; Yue et al., 2023). Predictive methods use convolutional networks to estimate blurring kernels and recover images but struggle with real-world conditions. To better handle real-world challenges, some approaches leverage Generative Adversarial Networks (GANs) (Goodfellow et al., 2014) to jointly learn the data distribution of the images and various degradation types. GAN-based methods significantly enhance

---

*Equal contribution.   †Corresponding author.

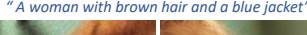
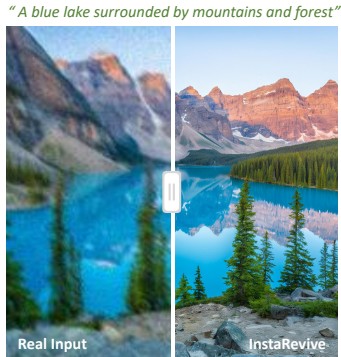
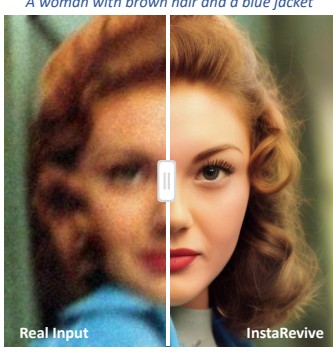
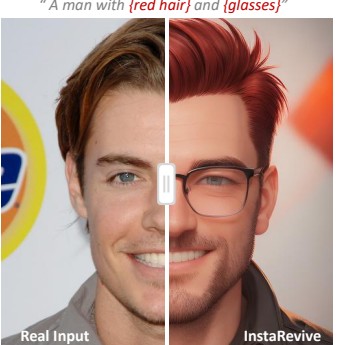

| (a) Blind image super-resolution | (b) Blind face restoration | (c) Face cartoonization |

Figure 1: Our InstaRevive showcases remarkable image enhancement capabilities across diverse tasks. Leveraging highly effective dynamic score matching with textual prompts, our framework adeptly harnesses the rich knowledge within the pre-trained diffusion model for (a) blind image super-resolution and (b) blind face restoration using **only one single step**. Furthermore, we demonstrate that InstaRevive seamlessly generalizes to additional related tasks such as (c) face cartoonization.

image quality but require careful tuning of sensitive hyper-parameters during training. Recently, diffusion models (DMs) (Ho et al., 2020; Rombach et al., 2022) have shown impressive visual generation capacity for image synthesis tasks. Some methods (Kawar et al., 2022; Wang et al., 2022a; Lin et al., 2023b; Yue et al., 2023) adopt the pre-trained diffusion models and restore the images during the denoising sampling. Despite their improvements in visual quality, these methods require lengthy and computationally intensive iterative inference.

To address the aforementioned challenges, we propose a novel diffusion distillation framework tailored for image enhancement. To maintain the image quality of diffusion models, we leverage the score-based distillation to align the HQ and generated data distributions. To accurately learn denoising trajectories, we introduce a dynamic control strategy for the diffusion noise scope, enabling the computation of precise scores and pseudo-GTs. This method not only refines distribution matching gradients but also overcomes the deficiency of conventional score matching, as depicted in Figure 2. After the dynamic score matching, our framework establishes a one-step mapping from low-quality inputs to high-quality results. This mapping is carefully optimized to ensure that the output images closely resemble the real-world HQ data distribution while distinctly diverging from undesirable distributions characterized by visible artifacts. To further exploit the generative priors embedded within the pre-trained text-to-image diffusion model, we use image captioning to extract natural language prompts, incorporating these as conditioning inputs for our framework.

We evaluate our framework on two representative image enhancement tasks: 1) blind face restoration (BFR) and 2) blind image super-resolution (BSR). Experimental results underscore the proficiency of our InstaRevive across these tasks. For BFR, InstaRevive achieves 22.3259 PSNR and 19.78 FID on the synthetic CelebA-Test. Additionally, it sets new benchmarks with a FID of 38.73 on the real-world LFW-Test. For BSR, we attain 0.4501 and 0.4722 MANIQA on the RealSet65 and RealSR datasets, demonstrating both high enhancement quality and efficiency in a single step. To further explore the potential of InstaRevive, we extend the applications of our framework to face cartoonization. InstaRevive consistently delivers visually appealing and plausible enhancements, underscoring the versatility and effectiveness of our framework, as illustrated in Figure 1.

## 2 RELATED WORKS

**Image enhancement.** Image enhancement involves tasks like denoising, deraining, and super-resolution, *etc*. Conventional works (Dong et al., 2014; Huang et al., 2020; Zhang et al., 2018a; Dong et al., 2015) utilize predictive models to estimate blur kernels and restore HQ images. With the rise of vision transformers (Dosovitskiy et al., 2020; Liu et al., 2021), some methods (Chen et al., 2021) incorporate the attention mechanism into basic architectures, yielding high-quality results. However, these models struggle with real-world degradations. The advent of generative models has introduced two main approaches in image enhancement, achieving significant success in complex blind image

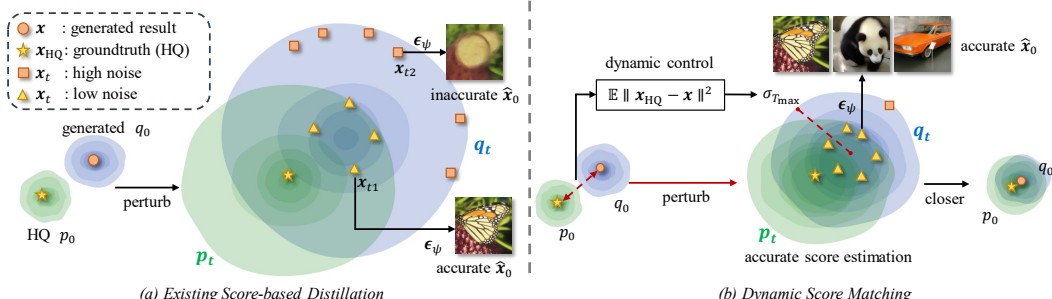

Figure 2: Comparison with existing score-based matching. (a) Existing score-based distillation uses a broad range of perturbations, causing large noise to shift the generated result $x$ far from the GT. This results in inaccurate score estimations (depicted by low-quality pseudo-GT $\hat{x}_0$) and impedes the distillation. (b) Our dynamic score matching adjusts $\sigma_{T_{\max}}$ to control the perturbation scale, ensuring more accurate score estimations and aligning distributions more closely.

restoration tasks. One approach is GAN-based methods (Wang et al., 2021a; Yuan et al., 2018; Fritsche et al., 2019; Zhang et al., 2021; Wang et al., 2021c;b; Yang et al., 2021; Zhou et al., 2022), which process images in the latent space to handle tasks like BSR and BFR. However, GAN-based methods require meticulous hyper-parameter tuning and they are often tailored to specific tasks, limiting their versatility. The other approach involves diffusion models (Ho et al., 2020; Rombach et al., 2022), known for their impressive image generation capabilities. Methods like (Wang et al., 2023c; Yue & Loy, 2022; Yue et al., 2023; Kawar et al., 2022; Fei et al., 2023; Lin et al., 2023b; Yu et al., 2024) design specific denoising structures to transfer image generation framework to image enhancement task. However, the sampling of diffusion models is time-consuming. To address this, some methods like (Xie et al., 2024; Wang et al., 2024; Zhu et al., 2024; Wu et al., 2024) employ distillation frameworks to process images in less steps. Nevertheless, distillation results often suffer from over-smoothing and reduced diversity, especially when facing complex degeneration.

**Diffusion distillation.** Diffusion distillation has gained attention for accelerating inference. Traditional diffusion acceleration methods such as (Zhao et al., 2023; Lu et al., 2022a;b) reduce sampling steps from 1000 to around 50 by solving ordinary differential equations (ODEs), significantly cutting inference time. Researchers have since introduced distillation techniques to reduce it to just a few or even one step. Methods like (Huang et al., 2023; Luhman & Luhman, 2021) use regression loss in pixel space for knowledge distillation. However, directly distilling the student model is challenging due to the complexity of predicting noise at each time step and the extensive training required on large-scale datasets. Progressive distillation (Meng et al., 2023; Salimans & Ho, 2022) address this by iteratively reducing inference steps. Inspired by these approaches, some methods (Liu et al., 2023; 2022; Luo et al., 2023; Song et al., 2023) aim to find a straighter inference path that shortens the sampling time to less than 5 steps. For one-step inference, (Yin et al., 2023; Nguyen & Tran, 2024) propose one-step frameworks using score-based distillation and produce commendable image quality. However, directly applying the score-based methods to image enhancement can result in inaccuracies and over-smoothing. To overcome this, our InstaRevive introduces dynamic control, which precisely learns the denoising trajectories and significantly improves enhancement results.

## 3 METHODOLOGY

In this section, we present the core concepts and detailed design of our InstaRevive. We will start with a brief introduction to diffusion models and score-based distillation. Subsequently, we will elucidate the motivation behind and the methodology of our dynamic score matching approach. Following this, we discuss the additional guidance with natural language prompts and describe the implementation of the framework. The overall pipeline of InstaRevive is depicted in Figure 3.

### 3.1 PRELIMINARIES

**Diffusion models.** Diffusion models are a family of generative models that can reconstruct the distribution of data by learning the reverse process of a diffusion process. In this process, noise

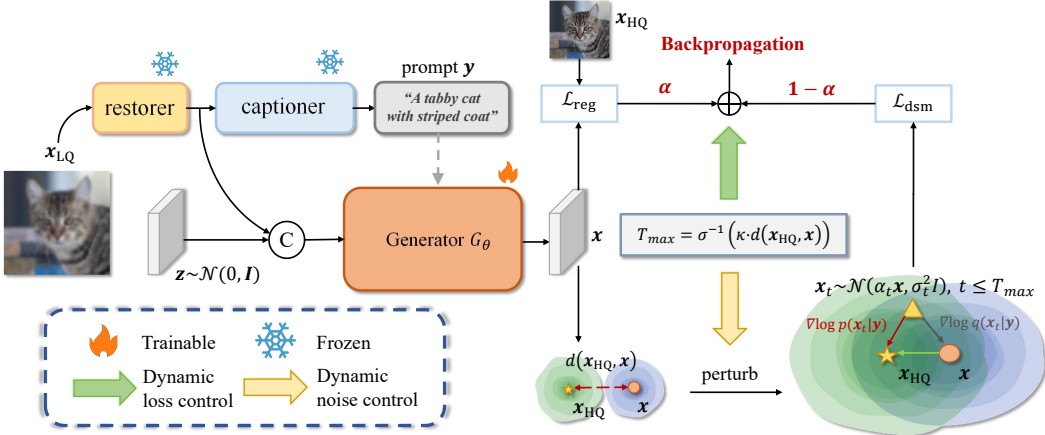

Figure 3: **The overall framework of *InstaRevive*.** InstaRevive utilizes a score-based diffusion distillation framework for image enhancement. During training, we employ two score estimators to calculate the gradients of the KL divergence. To improve estimation accuracy, we devise a dynamic control strategy to regulate the diffusing scope and adjust loss function weights. During inference, our generator can yield high-quality and visually appealing results in a single step.

is incrementally added to data, while the reverse process predicts the distribution of progressively denoised data points. Given a random noise $\epsilon \sim \mathcal{N}(0, \boldsymbol{I})$, the diffusion process is defined as:

$$\boldsymbol{x}_t = \alpha_t \boldsymbol{x}_0 + \sigma_t \epsilon, \tag{1}$$

where $\boldsymbol{x}_0$ and $\boldsymbol{x}_t$ are the clean and noisy image, and $\{(\alpha_t, \sigma_t)\}_{t=1}^T$ is the noise schedule. Conversely, the reverse process starts from a pure noise $\boldsymbol{x}_T \sim \mathcal{N}(0, \boldsymbol{I})$, and a denoising model $\epsilon_\psi$ iteratively predicts the noise, transitioning from $\boldsymbol{x}_{t+1}$ to $\boldsymbol{x}_t$. Specifically, in text-to-image diffusion models, the text prompt $\boldsymbol{y}$ serves as guidance during each denoising step. The training objective is to minimize:

$$\mathcal{L}_{\text{DM}} = \mathbb{E}_{\boldsymbol{x}_0, \epsilon, t} \left\| \epsilon - \epsilon_\psi(\boldsymbol{x}_t, t, \boldsymbol{y}) \right\|_2^2. \tag{2}$$

After training, the diffusion model learns the gradient of data density $\nabla_{\boldsymbol{x}_t} \log p_t(\boldsymbol{x}_t|\boldsymbol{y})$ via the noise prediction network $\epsilon_\psi$. We denote this gradient as the *score function* of $p_t$ and it can be approximated by $\nabla_{\boldsymbol{x}_t} \log p_t(\boldsymbol{x}_t|\boldsymbol{y}) \approx -\epsilon_\psi(\boldsymbol{x}_t, t, \boldsymbol{y})/\sigma_t$.

**Score-based distillation.** The score-based distillation (Poole et al., 2022; Wang et al., 2023a; Lin et al., 2023a) was firstly introduced in text-to-3D generation. Given a 3D representation $\theta$ (*e.g.* NeRF) and a rendering function $g(\cdot, c)$, the rendered image can be obtained by $\boldsymbol{x}_0 = g(\theta, c)$ with a specific camera position $c$. The objective is to optimize this representation $\theta$ such that its 2D-rendered images align with the pre-trained diffusion model's outputs given a text prompt. However, directly addressing this optimization problem is challenging due to the complexity of the diffusion model's distribution, often resulting in issues such as over-saturation or over-smoothing. To tackle this problem, (Wang et al., 2023d) proposes Variational Score Distillation (VSD) by introducing an additional score function estimated by another model $\epsilon_\phi(\boldsymbol{x}_t, t, \boldsymbol{y}, c)$, which can be finetuned with LoRA (Hu et al., 2021) during optimization with the diffusion loss as described in Equation 2. The final optimization objective of VSD is derived as:

$$\min_{\mu} \mathbb{E}_{t, c} \left[ (\sigma_t/\alpha_t) \omega(t) D_{\text{KL}}(q_t^\mu(\boldsymbol{x}_t|c, \boldsymbol{y}) || p_t(\boldsymbol{x}_t|\boldsymbol{y}^c)) \right], \tag{3}$$

where $t \sim \mathcal{U}(T_{\min}, T_{\max})$, $w(t)$ is a weighting function, $\boldsymbol{y}^c$ is a view-specific prompt, and $\mu(\theta|\boldsymbol{y})$ is the probabilistic density of $\theta$ given the prompt.

## 3.2 Dynamic Score Matching for Image Enhancement

**Simplify the objective of image enhancement.** The primary objective in optimizing image enhancement tasks typically involves a regression term that penalizes the distance between the generated result and the ground truth. However, relying solely on the regression term $\mathcal{L}_{reg}$ to train the generator $G_\theta$ often yields unsatisfactory results. This is due to the complex and often irreversible degradation

process $\boldsymbol{x}_{\mathrm{LQ}} = D(\boldsymbol{x}_{\mathrm{HQ}})$ encountered in real-world scenarios, which significantly impairs image information. Under such challenging degradation conditions, it is difficult to construct a one-to-one mapping between HQ and LQ images with only a simple regression approach. To ease this challenge, we relax the optimization with another term that penalizes the distance between the distributions of generated results and HQ images. Inspired by the score-based distillation, we formulated this term as the KL divergence, and the optimization objective is revised as follows:

$$\min_{\theta} \mathbb{E}_{\boldsymbol{x}_{\mathrm{LQ}} \sim \mathcal{X}} \|G_{\theta}(\boldsymbol{x}_{\mathrm{LQ}}) - \boldsymbol{x}_{\mathrm{HQ}}\|_2^2 + \lambda D_{\mathrm{KL}} \left( p_0 \| q_0 \right), \tag{4}$$

where $\mathcal{X}$ represents the set of LQ images, $p_0$ and $q_0$ denote the distribution of HQ images and generated results, and $\lambda$ is the weight factor. By minimizing these two terms concurrently, we alleviate the learning difficulty and improve result diversity. The generated result $\boldsymbol{x}$ is not strictly forced to match $\boldsymbol{x}_{\mathrm{HQ}}$, but rather to fall within the HQ distribution, allowing for various plausible results that reduce penalties. This additional term also acts as a regularization, preventing overfitting and bolstering generator robustness. Unlike GANs, which train a discriminator and only provide the probability of realness, our approach aims to expand the gradient of this term as score tensors that explicitly point out the updating direction. Given the robust gradient estimation and extensive knowledge of HQ images inherent in diffusion models, we leverage these models to compute the gradient of these two distributions and optimize our generator through diffusion distillation.

**Score matching distillation framework.** To solve the optimization problem in Equation 4, we aim to diminish the KL divergence term $D_{\mathrm{KL}}(q_0 \| p_0)$. Given the prompts $\boldsymbol{y}$, we condition the probability distributions on $\boldsymbol{y}$. However, $p_0(\boldsymbol{x}|\boldsymbol{y})$ vanishes when $\boldsymbol{x}$ is far away from HQ images, which makes the training difficult to converge effectively. In response, Score-SDE (Song & Ermon, 2019; Song et al., 2020) introduces a method that diffuses the original distributions with varying scales of noise indexed by $t$ and optimizes a series of KL divergences between these diffused distributions, $D_{\mathrm{KL}}(p_t \| q_t)$, the gradient of which is:

$$
\begin{aligned}
\nabla_{\theta} D_{\mathrm{KL}} &= \mathbb{E}_{t, \boldsymbol{\epsilon}, \boldsymbol{x}_{\mathrm{LQ}}} \left[ -\omega(t) \left( \nabla_{\boldsymbol{x}_t} \log p_t(\boldsymbol{x}_t|\boldsymbol{y}) - \nabla_{\boldsymbol{x}_t} \log q_t(\boldsymbol{x}_t|\boldsymbol{y}) \right) \frac{\partial G_{\theta}(\boldsymbol{x}_{\mathrm{LQ}}, \boldsymbol{y})}{\partial \theta} \right] \\
&= \mathbb{E}_{t, \boldsymbol{\epsilon}, \boldsymbol{x}_{\mathrm{LQ}}} \left[ \sigma_t \omega(t) \left( \boldsymbol{\epsilon}_{\psi}(\boldsymbol{x}_t, \boldsymbol{y}) - \boldsymbol{\epsilon}_{\phi}(\boldsymbol{x}_t, \boldsymbol{y}) \right) \frac{\partial G_{\theta}(\boldsymbol{x}_{\mathrm{LQ}}, \boldsymbol{y})}{\partial \theta} \right],
\end{aligned}
\tag{5}
$$

where $\boldsymbol{x}_t$ is the noisy image obtained by Equation 1, $p_t$ and $q_t$ are the diffused distributions, and $\omega(t)$ is the weight related to timestep $t$. (For detailed derivations, please refer to A.8.1.) The scores of the diffused distributions can be estimated using two diffusion models, $\boldsymbol{\epsilon}_{\psi}$ and $\boldsymbol{\epsilon}_{\phi}$, as mentioned earlier. With increasing $t$ from 0 to $T$, the overlap between the diffused distributions grows, making the scores well-defined and facilitating the calculation via the diffusion models. During training, we backpropagate the gradient in Equation 5 to update the generator. Simultaneously, we update the diffusion model $\boldsymbol{\epsilon}_{\phi}$ to align with the generated distribution using the diffusion loss in Equation 2.

**Dynamic noise control for effective matching.** The accuracy of Equation 5 depends on the precise estimation of gradients by the two score functions, $\nabla_{\boldsymbol{x}_t} \log p_t(\boldsymbol{x}_t|\boldsymbol{y})$ and $\nabla_{\boldsymbol{x}_t} \log q_t(\boldsymbol{x}_t|\boldsymbol{y})$. In an ideal scenario, the diffusion models accurately compute these scores. However, during actual training, we observe that the estimates from the diffusion models might deviate, leading to potential inaccuracies, especially as $t$ nears $T_{\max}$. As shown in Figure 2, this discrepancy can yield incorrect gradients for the KL divergence, impeding effective parameter updates and hindering training progress. To overcome this challenge, we propose the dynamic noise control strategy. Unlike the typical application scenarios like text-to-image generation, where the generated image distribution diverges significantly from the target distribution initially, in image enhancement, the generated distribution closely approximates the high-quality distribution from the outset. Hence, we employ relatively subtle Gaussian noise perturbations to the data distributions, ensuring the rationality of the two score functions. Furthermore, this controlled noise diminishes the gap between $\boldsymbol{x}_t$ and $\boldsymbol{x}_{\mathrm{GT}}$, contributing to accurate score estimates. Specifically, we regulate the added Gaussian noise level as follows:

$$T_{\max} = \sigma^{-1}[\kappa \cdot d(\boldsymbol{x}_{\mathrm{HQ}}, \boldsymbol{x})], \quad t \sim \mathcal{U}[0.02 T_{\max}, T_{\max}] \tag{6}$$

$$d(\boldsymbol{x}_{\mathrm{HQ}}, \boldsymbol{x}) = \sqrt{\Sigma_{i=1}^{B} \|\boldsymbol{x}_{\mathrm{HQ}}^{(i)} - \boldsymbol{x}^{(i)}\|_2^2 / B} \tag{7}$$

where $\sigma^{-1}$ is the inverse function of $\sigma_t$, $\kappa$ is the control factor and $B$ is the batch size. By adjusting $T_{\max}$, we assure that $\mathrm{Var}(\boldsymbol{x}_t - \boldsymbol{x}) \leq \mathrm{Var}(\boldsymbol{x}_{T_{\max}} - \boldsymbol{x}) = \sigma_{T_{\max}}^2 \boldsymbol{I}$, controlling the distance between

$x_t$ and $x_{\mathrm{HQ}}$ via the triangle inequality (elaborated in Equation 15), which ensures more accurate score computation at $x_t$. Additionally, our score matching strategy is dynamic, allowing the generator to focus on varying noise levels during different training stages. We denote the score matching loss governed by this dynamic scheme as $\mathcal{L}_{dsm}$. Specifically, Equation 6 results in different $T_{\max}$ with respect to the current discrepancy between HQ and generated distribution. As training advances and these two distribution data become more aligned, a smaller $T_{\max}$ guides the generator to concentrate mainly on refining detailed artifacts in generated images.

**Dynamic loss control for stable training.** Next, we compare the regression loss and dynamic score matching loss. Regression loss, commonly used in image restoration, penalizes pixel-level errors but may miss perceptual quality. In contrast, dynamic score matching loss evaluates distribution similarity between real and generated data, enhancing realism and texture, but lacking pixel-level guidance. We employ two diffusion models $\epsilon_\psi$ and $\epsilon_\phi$, to estimate the gradients of the real and generated data distribution, respectively. Both are initialized from pre-trained image generation diffusion models. While $\epsilon_\psi$ accurately estimates the gradient of the real data distribution, the gradient estimated by $\epsilon_\phi$ is significantly less accurate. Consequently, Equation 5 fails to provide proper optimization directions for minimizing the KL divergence, potentially hindering generator training or even introducing negative effects. To balance these loss functions, we propose a dynamic control strategy. Early in training, the regression loss drives effective optimization, as dynamic score matching is biased. As the generator improves, regression loss becomes less informative, and score matching focuses on refining finer details. In deployment, we use $\alpha$ to control the ratio between the two loss functions, which is obtained by linearly mapping $T_{\max}$ to the interval $[0, 1]$.

### 3.3 GUIDANCE WITH NATURAL LANGUAGE PROMPTS

Leveraging the pre-trained diffusion model's knowledge of the relationship between textual and visual information, we introduce additional textual guidance to further improve image processing capabilities. Unlike implicit visual features from LQ images, textual prompts provide explicit and detailed descriptions, significantly aiding in our image enhancement. This is especially valuable for severely degraded images, where visual perception alone may be ambiguous. To enable the captioner to handle extremely low-quality images, we employ a pre-trained image restorer for coarse restoration. Compared to the model backbone, the restorer holds much fewer parameters and exerts minimal impact on inference speed. By incorporating proper textual prompts, we can set clear goals for the enhancement result and generate reasonable outcomes from chaotic LQ images. This approach also allows for creative and editable results, offering high controllability through human guidance, enhancing restoration performance and extending the framework's flexibility.

### 3.4 IMPLEMENTATION

We maintain a consistent framework structure across all tasks, employing the pre-trained diffusion model to initialize the student and teacher models. This approach exploit the rich pre-trained knowledge of image understanding and vision-language relationships acquired during the generation task. Specifically, for the generator $G_\theta$ and two score estimators, $\epsilon_\psi$ and $\epsilon_\phi$, we initialize them with identical weights from the pre-trained text-to-image diffusion model. To estimate the generated distribution, we unfreeze the parameters of the score estimator $\epsilon_\phi$. For the restorer, we employ the lightweight module in (Liang et al., 2021) following (Lin et al., 2023b) to perform a coarse restoration. To gather textual prompts, we utilize the multi-modal BLIP (Li et al., 2022a) as the image captioner, extracting semantic contents in the images. To enhance the diversity, we concatenate $x_{\mathrm{LQ}}$ with random noise $z$ and use a convolution layer to align the channel dimension. To approximate the degradation conditions in BFR and BSR, we produce the synthetic data from HQ image by $x_{\mathrm{LQ}} = [(k * x_{\mathrm{HQ}}) \downarrow_r + n]_{\mathrm{JPEG}}$, which consists of blur, noise, resize and JPEG compression.

## 4 EXPERIMENTS

### 4.1 EXPERIMENT SETUPS

**Datasets.** For blind face restoration (BFR), we utilize the Flickr-Faces-HQ (FFHQ) (Karras et al., 2019), which encompasses 70,000 high-resolution images. We resize them to $512 \times 512$ to match

Table 1: **Quantitative comparisons for BFR on the synthetic and real-world datasets.** We highlight the best and second best performance in **bold** and underline, respectively. We categorize the methods into conventional (up), diffusion-based (middle) and distilled (bottom). Our InstaRevive shows very competitive results compared with existing methods. We obtain remarkable image quality and identity consistency with the leading FID and PSNR scores. Our framework also exhibits much faster inference than the diffusion-based method.

| Method | Synthetic Dataset CelebA | | | | | Wild Datasets | | FPS↑ |
| | PSNR↑ | SSIM↑ | LPIPS↓ | FID↓ | IDS ↑ | LFW FID↓ | WIDER FID↓ | |
|---|---|---|---|---|---|---|---|---|
| GPEN (Yang et al., 2021) | 21.3995 | 0.5742 | 0.4687 | 23.92 | 0.48 | 51.97 | 46.35 | 7.278 |
| GCFSR (He et al., 2022) | 21.8791 | 0.6072 | 0.4577 | 35.49 | 0.44 | 52.20 | 40.86 | 9.243 |
| GFPGAN (Wang et al., 2021b) | 21.6953 | 0.6060 | 0.4304 | 21.69 | 0.49 | 52.11 | 41.70 | 8.152 |
| VQFR (Gu et al., 2022) | 21.3014 | **0.6132** | 0.4116 | 20.30 | 0.48 | 49.88 | 37.87 | 3.837 |
| RestoreFormer (Wang et al., 2022c) | 21.0025 | 0.5283 | 0.4789 | 43.77 | 0.56 | 48.43 | 49.79 | 4.964 |
| DMDNet (Li et al., 2022b) | 21.6617 | 0.6000 | 0.4828 | 64.79 | **0.67** | 43.36 | 40.51 | 3.454 |
| CodeFormer (Zhou et al., 2022) | 22.1519 | 0.5948 | 0.4055 | 22.19 | 0.47 | 52.37 | 38.78 | 5.188 |
| DifFace-100 (Yue & Loy, 2022) | 22.1483 | 0.6057 | 0.4129 | 19.95 | 0.61 | 46.17 | 37.42 | 0.225 |
| ResShift-4 (Yue et al., 2023) | 21.6858 | 0.5829 | 0.4082 | 20.03 | 0.59 | 50.09 | 37.21 | 3.623 |
| InstaRevive (Ours) | **22.3259** | 0.6109 | **0.4025** | **19.78** | 0.65 | **38.73** | **35.29** | 7.967 |

the input scale of the diffusion model. For evaluation, we leverage the widely used CelebA-Test dataset (Liu et al., 2015), which consists of 3,000 synthetic HQ-LQ pairs. Additionally, to further validate the effectiveness on real-world data, we employ 2 wild face datasets, LFW-Test (Wang et al., 2021b) and WIDER-Test (Zhou et al., 2022) which contain face images with varying degrees of degradation. For blind image super-resolution, we train our framework using the large-scale ImageNet dataset (Deng et al., 2009) and evaluate on the RealSR (Cai et al., 2019) and RealSet65 (Yue et al., 2023). Specifically, RealSR contains 100 LQ images captured by two different cameras in diverse scenarios, while RealSet65 comprises 65 LQ images sourced from widely used datasets and the Internet. For face cartoonization, we create a stylized dataset using the iterative sampling of ControlNet (Zhang et al., 2023). This dataset includes 130,000 $512 \times 512$ images based on FFHQ.

**Training details.** We unfreeze two models in our framework during training: the generator $G_\theta$ and the score estimator $\epsilon_\phi$ for the generated distribution. Both models are optimized with a batch size of 32 and a learning rate of 1e-6 using two AdamW optimizers with a weight decay of 1e-2. We initialize the generator and two score estimators by replicating the denoising transformer blocks in (Chen et al., 2023a). For BFR and BSR, we employ the high-order degradation model in (Wang et al., 2021c), training for 25K and 35K steps with 4 Nvidia A800 GPUs, respectively. For face cartoonization, we finetune our pre-trained BFR model for an additional 10K steps. We set the KL term weight to 1.0 and the control factor to 1.5 for optimal performance.

**Metrics.** To evaluate our InstaRevive's performance on BFR, we calculate three traditional metrics, including PSNR, SSIM and LPIPS (Zhang et al., 2018b), on the CelebA-Test dataset. However, these metrics have their limitations in assessing visual quality as they often penalize high-frequence details in our generated images, *e.g.*, hair texture. Therefore, we also include the widely-used FID (Heusel et al., 2017) score to measure overall image quality. Additionally, we compute the identity similarity (IDS) using a pre-trained face perception network (Deng et al., 2019). For BSR, we leverage four non-reference metrics MANIQA (Yang et al., 2022), MUSIQ (Ke et al., 2021), CLIPIQA (Wang et al., 2023b) and NIQE to assess image quality. To demonstrate the efficiency of InstaRevive, we also compare the throughput (FPS) of our framework with other methods.

## 4.2 MAIN RESULTS

**Blind face restoration.** Blind face restoration necessitates a convincing mapping from LQ to the desired face image with high-quality details. We evaluate InstaRevive on both synthetic CelebA-Test (Liu et al., 2015), with in-the-wild LFW-Test (Wang et al., 2021b) and WIDER-Test (Zhou et al., 2022). Our comparative analysis involves recent state-of-the-art methods, including GPEN (Yang et al., 2021), GCFSR (He et al., 2022), GFPGAN (Wang et al., 2021b), VQFR (Gu et al., 2022), RestoreFormer (Wang et al., 2022c), DMDNet (Li et al., 2022b), CodeFormer (Zhou et al., 2022), DifFace (Yue & Loy, 2022) (100 steps) and ResShift (Yue et al., 2023) (4 steps). As depicted in Table 1, our InstaRevive achieves notable performance metrics, including 19.78 FID, 22.3258 PSNR

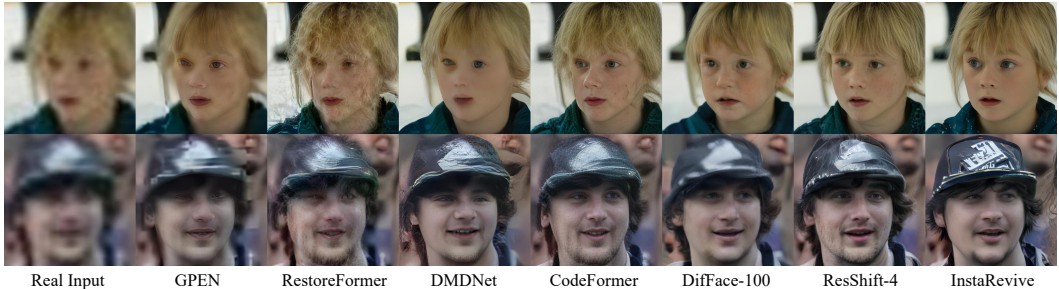

Real Input   GPEN   RestoreFormer   DMDNet   CodeFormer   DifFace-100   ResShift-4   InstaRevive

Figure 4: **Qualitative comparisons on the real-world faces.** Our method demonstrates impressive enhancement capabilities on real-world faces, producing high-fidelity and visually appealing faces. Compared to other methods, InstaRevive exhibits robustness when handling challenging cases.

Table 2: **Quantitative comparisons for BSR on real-world datasets.** Our framework achieves high-quality enhancement and outperforms existing methods in various image quality metrics.

| Type | Method | RealSet65 | | | | RealSR | | | |
|------|--------|-----------|--|--|--|--------|--|--|--|
| | | MANIQA↑ | MUSIQ↑ | NIQE↓ | CLIPIQA↑ | MANIQA↑ | MUSIQ↑ | NIQE↓ | CLIPIQA↑ |
| GAN | RealSR-JPEG (Ji et al., 2020) | 0.2923 | 50.537 | 4.8042 | 0.5280 | 0.1738 | 36.071 | 6.9528 | 0.3614 |
| | RealESRGAN (Wang et al., 2021c) | 0.3064 | 42.366 | 4.8909 | 0.3737 | 0.2037 | 29.034 | 7.7273 | 0.2363 |
| | BSRGAN (Zhang et al., 2021) | 0.3876 | 65.578 | 5.5852 | 0.6160 | 0.3705 | 63.584 | 4.6606 | 0.5434 |
| | SwinIR-GAN (Liang et al., 2021) | 0.2699 | 44.975 | 8.0458 | 0.4225 | 0.2727 | 43.219 | 7.7362 | 0.3637 |
| Diffusion | StableSR (Wang et al., 2023c) | 0.3643 | 59.670 | 4.8932 | 0.5633 | 0.3807 | 61.362 | 4.6778 | 0.5530 |
| | GDP (Fei et al., 2023) | 0.2757 | 55.874 | 6.8496 | 0.6339 | 0.2865 | 59.378 | 7.0729 | 0.6533 |
| | ResShift-15 (Yue et al., 2023) | 0.3958 | 61.330 | 5.9425 | 0.6537 | 0.3640 | 59.873 | 5.9820 | 0.5958 |
| | LDM-15 (Rombach et al., 2022) | 0.2760 | 47.600 | 6.2490 | 0.4203 | 0.2909 | 50.926 | 5.9172 | 0.3932 |
| Distilled | SinSR (Wang et al., 2024) | 0.4374 | 62.635 | 5.9675 | **0.7138** | 0.4015 | 59.278 | 6.2683 | **0.6638** |
| | InstaRevive (Ours) | **0.4571** | **65.849** | **4.1995** | 0.6755 | **0.4722** | **64.535** | **4.2781** | 0.6577 |

and 0.4025 LPIPS, outperforming other methods. We also obtain comparable SSIM and IDS and establish a higher upper bound of the diffusion-based methods. Furthermore, we achieve 38.73 FID on the LFW-Test, demonstrating high restoration quality for real-world images. We also attain competitive performance on WIDER-Test. Besides, our InstaRevive demonstrates rapid processing with a 7.967 FPS, significantly outpacing the diffusion-based methods. The qualitative results in Figure 4 further illustrate that our method effectively restores face images degraded by real-world conditions, consistently delivering visually appealing outcomes with single-step inference.

**Blind image super-resolution.** Blind image super-resolution involves general image prior and low-level structural knowledge. We evaluate our InstaRevive on RealSR (Cai et al., 2019) and RealSet65 (Yue et al., 2023), comparing it with cutting-edge methods, including GAN-based RealESR-GAN (Wang et al., 2021c), BSRGAN (Zhang et al., 2021), SwinIR-GAN (Liang et al., 2021), and diffusion-based GDP (Fei et al., 2023), ResShift (Yue et al., 2023), LDM (Rombach et al., 2022) and StableSR (Wang et al., 2023c). As shown in Table 2, our InstaRevive achieves outstanding image quality with the highest MANIQA, MUSIQ and NIQE on both datasets. Note that our one-step generator outperforms other methods that use iterative sampling by a significant margin on NIQE. Additionally, we attain competitive CLIPIQA scores compared with the most recent method employing diffusion distillation. We also provide qualitative comparison in Figure 5 with recent multi-step models like DiffBIR (Lin et al., 2023b). As illustrated, our generator produces high-quality enhancements and plausible details, despite utilizing only a single inference step, demonstrating an excellent balance between efficiency and performance. Further analysis of the model parameters and inference time is discussed in Section A.4. Moreover, on the synthetic ImageNet-Test dataset, our method achieves impressive PSNR and SSIM metrics, as reported in Table 4 in the appendix.

## 4.3 ANALYSIS

**Effective distillation of diffusion priors.** The major challenge in adapting diffusion distillation to image enhancement is converting the generative capacity into restoration power within a single-step model. To demonstrate our framework enables effective diffusion distillation, we perform direct distillation using only regression loss to supervise the generator's outputs. Results in Table 3 and Figure 6 indicate that this approach (w/o score) yields blurry outcomes with visible artifacts. In contrast, our score matching approach, by minimizing the KL divergence, updates the student model in a "softer" and more effective way, thereby reducing the distance between the HQ image and the

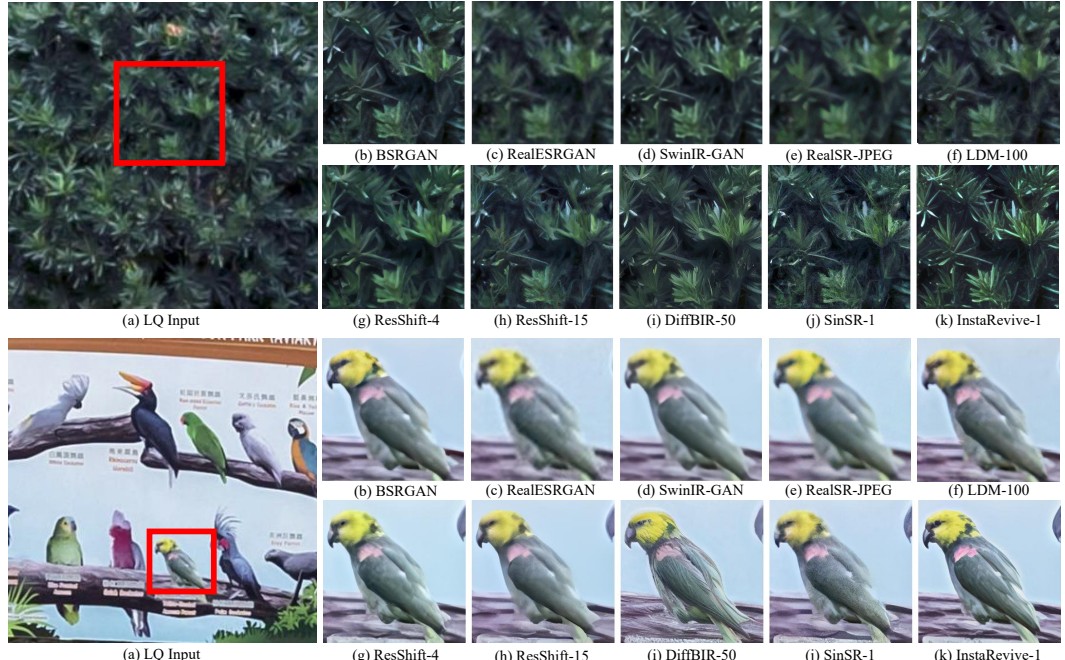

Figure 5: **Qualitative comparisons on real-world datasets.** Our InstaRevive delivers exceptional details with just one-step inference. The numbers following each method indicate the corresponding inference steps. More results and comparisons can be found in Figure. 9 and Figure. 10.

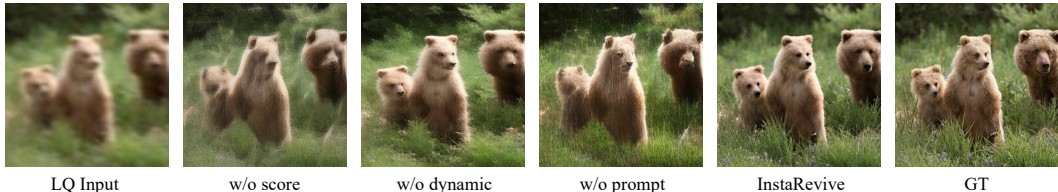

Figure 6: **Visual results of the ablations.** Our dynamic score matching and prompt guidance significantly enhance both image quality and controllability in the generated outputs.

current generated results. Unlike the one-to-one mapping, the KL divergence term allows for a range of plausible outcomes, imparting more detailed and comprehensive knowledge to the student model.

**Dynamic control for accurate learning.** To evaluate the dynamic noise and loss control strategy, we conducted an ablation study by removing it while keeping other hyper-parameters fixed. The results (w/o dynamic), presented in Table 3 and Figure 6, demonstrate that the dynamic control strategy greatly improves the quality of the generated images. Furthermore, the loss curves in Figure 7 indicate that this approach accelerates training and shortens convergence time significantly. This highlights the importance of maintaining the noise level within an appropriate range during the diffusion process and dynamically adjusting the loss weights. This dynamic control strategy is particularly well-suited for enhancement tasks, where the distributions of the generated and real images initially overlap, necessitating a focus on the noisy image not too far away from $x_{HQ}$, thus reducing the training difficulty and enhancing the model performance.

**Additional guidance with textual prompts.** Textual prompts offer clear and explicit guidance for the enhancement target. To evaluate their significance, we perform an ablation study using null textual input for training and evaluation. As shown in Table 3 and Figure 6, the absence of textual prompts (w/o prompt) results in a drop in performance, especially in more challenging scenarios. This suggests that textual prompts are crucial for the teacher model $\epsilon_\psi$ to generate the correct scores.

**Limitations.** Despite InstaRevive's promising results within a one-step paradigm, it may produce suboptimal results when encountering extreme degradation or complex content, as shown in Figure 19. How to utilize the generative power to address these challenges remains an area for future exploration.

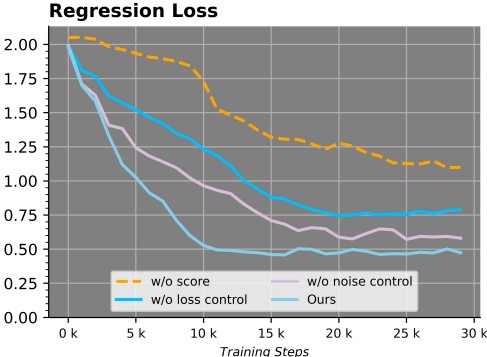

Figure 7: **Loss curves of the ablations.** Our design promotes faster convergence.

Table 3: **Ablation studies.** We perform ablations on BFR and BSR to validate the effectiveness of InstaRevive's components. The results demonstrate that score matching leads to effective distillation and that dynamic control and textual prompts are beneficial for overall performance.

| Method | FID↓ | | MANIQA↑ | |
|---|---|---|---|---|
| | CelebA | LFW | RealSet65 | RealSR |
| w/o score | 26.45 | 51.03 | 0.4085 | 0.4259 |
| w/o dynamic | 22.43 | 45.39 | 0.4287 | 0.4463 |
| w/o prompt | 19.90 | 39.38 | 0.4374 | 0.4541 |
| InstaRevive (Ours) | **19.78** | **38.73** | **0.4571** | **0.4722** |

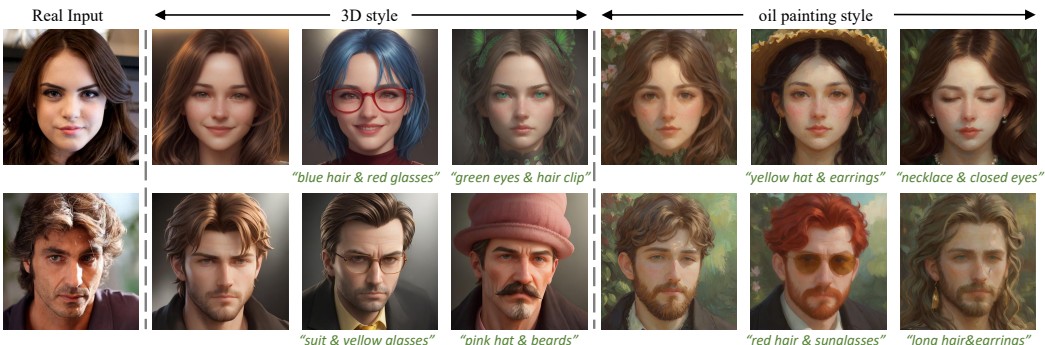

Figure 8: **Qualitative results on face cartoonization.** InstaRevive delivers high-quality results (Col.2 and 5). Furthermore, we can edit the details with textual prompts (Col.3, 4, 6 and 7), showcasing high controllability. More comparisons can be found in Figure 13.

## 4.4 EXTENSIONS

To showcase InstaRevive's versatility and controllability, we extend its application to the complex task of controllable face cartoonization. This task entails not only altering the global style of the images but also editing their detailed content based on the natural language prompts. After fine-tuning our framework on the stylized face dataset, InstaRevive learns to edit the style and the semantic regions from the pre-trained diffusion model and textual prompts. As illustrated in Figure 8, our generator produces high-quality results with remarkable controllability. These results also affirm InstaRevive's potential for broader applications in image editing and enhancement tasks. To ensure high *identity consistency* during image-to-image transitions, we utilize the IP-Adapter (Ye et al., 2023) as our teacher model. The details and results are presented in Section. A.5. We further explore more tasks including *low-light enhancement* and *face inpainting*, with findings detailed in Section. A.3.

## 5 CONCLUSION

We propose InstaRevive, a one-step image enhancement framework that excels in efficiency and effectiveness across various tasks. Empowered by the dynamic control strategy for score matching distillation and additional guidance with textual prompts, InstaRevive enables accurate computation of gradients for both HQ and generated data distributions, thereby significantly accelerating the training process while improving the result quality in only one-step inference. We also extend InstaRevive to face cartoonization, showcasing its strong generalization. We hope our attempt can inspire future work to further exploit diffusion priors for image enhancement.

**Acknowledgment:** This work was supported in part by the National Key Research and Development Program of China under Grant 2023YFB280690, and in part by the National Natural Science Foundation of China under Grant Grant 62125603, Grant 62321005, Grant 62336004 and Grant 62306031.

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

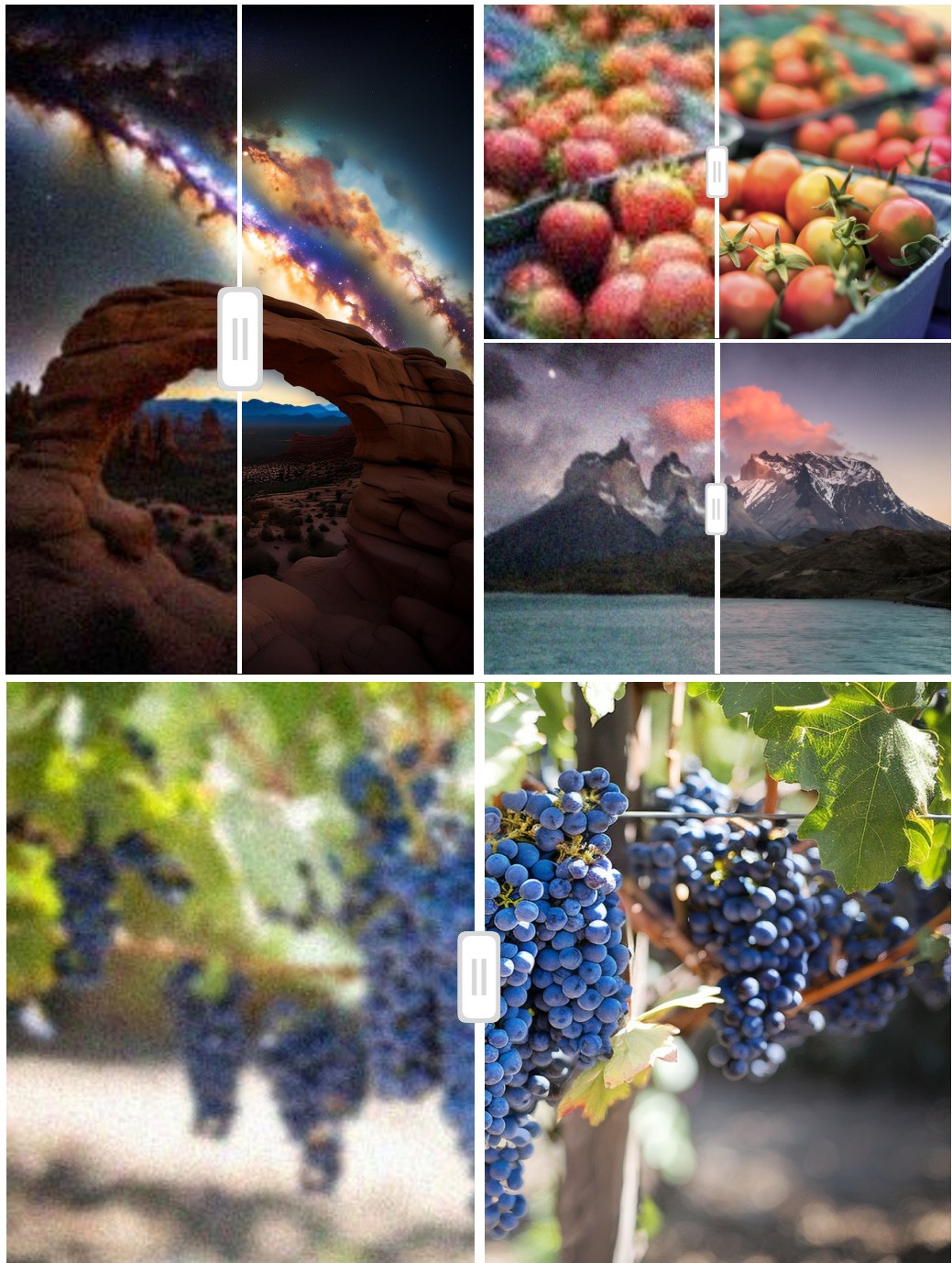

Figure 9: **More qualitative results on real-world images with larger resolution.** InstaRevive excels at enhancing real-world images with high resolutions, such as 2K. This figure contrasts the input (left) with InstaRevive's output (right), highlighting its impressive visual performance.

## A    APPENDIX

### A.1    MORE QUALITATIVE RESULTS

We extend our framework to accommodate higher-resolution images with diverse aspect ratios. As illustrated in Figure 9, we provide more qualitative results on larger-scale images, demonstrating our model's ability to manage high-resolution images and restore pixel-level details. For further

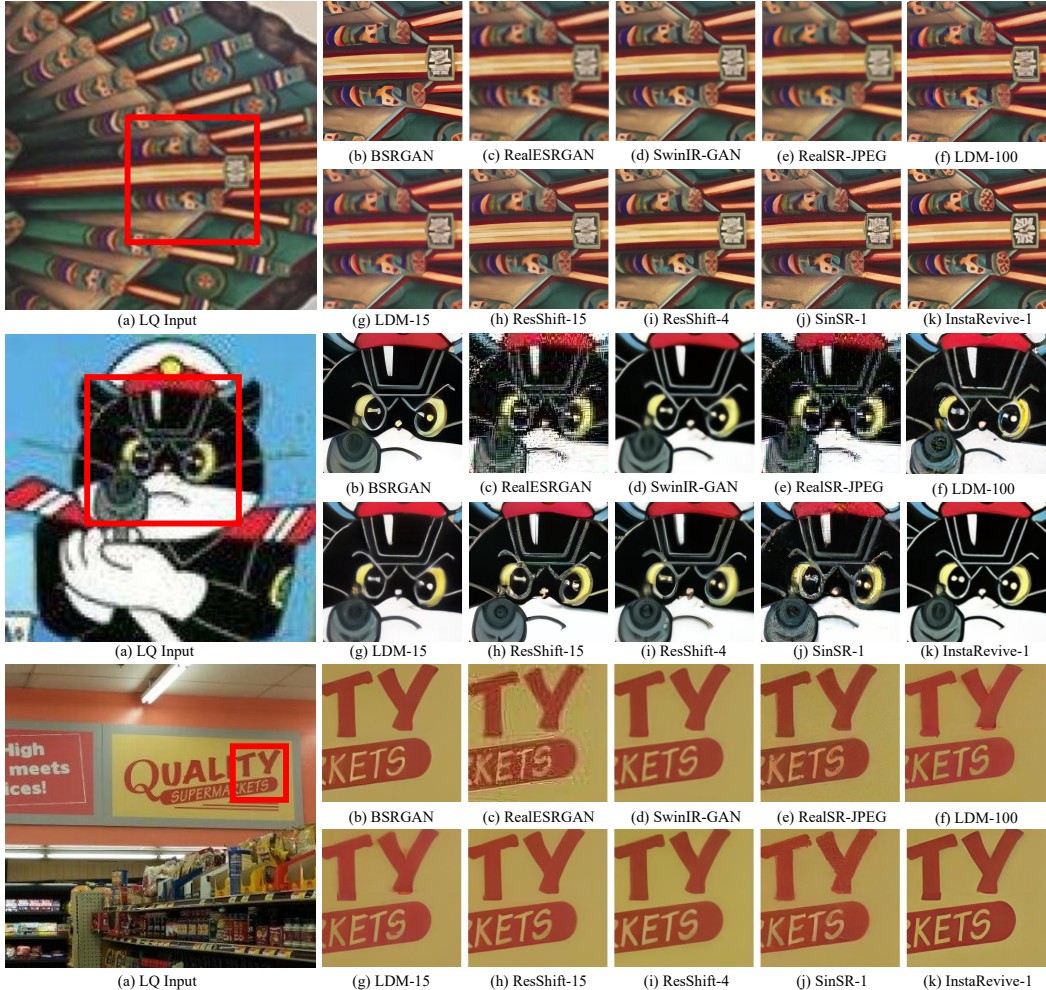

Figure 10: **More qualitative comparisons on real-world datasets.** InstaRevive demonstrates high-performance blind image super-resolution on real-world images, consistently producing clear and detailed results with only a single inference step.

evaluation, we compare our method with recent state-of-the-art approaches, including OSEDiff (Wu et al., 2024) and SUPIR (Yu et al., 2024), in Figure 12. The results reveal that our framework delivers highly competitive performance. Moreover, we showcase more comparisons on the BSR task in Figure 10, highlighting InstaRevive's versatility in enhancing styles ranging from comic images to real photos. We also provide more qualitative results of BFR tasks in Figure 11. Our InstaRevive successfully restores real-world images with challenging degradations, demonstrating the efficacy and robustness of our framework. For face cartoonization, we compare our InstaRevive with current diffusion-based methods like SDEdit (Meng et al., 2021) (40 steps) and InstuctPix2Pix (Brooks et al., 2023) (100 steps). As shown in Figure 13, our framework yields visually appealing outcomes with finer details. Note that our framework only requires one step for inference.

## A.2 QUANTITATIVE RESULT ON SYNTHETIC DATASET

To further assess the consistency of results in the BSR task, we conduct evaluations on the synthetic ImageNet-Test dataset, as proposed in ResShift (Yue et al., 2023). This dataset comprises 3,000 images randomly selected from the ImageNet validation set based on the widely-used degradation model. The results, presented in Table 4, indicate that our method achieves highly competitive consistency metrics, such as PSNR and SSIM. Additionally, we report the performance of SwinIR-GAN (Liang et al., 2021), which serves as the restorer in our framework. The results reveal that the restorer struggles to handle complex degradation scenarios, resulting in subpar performance.

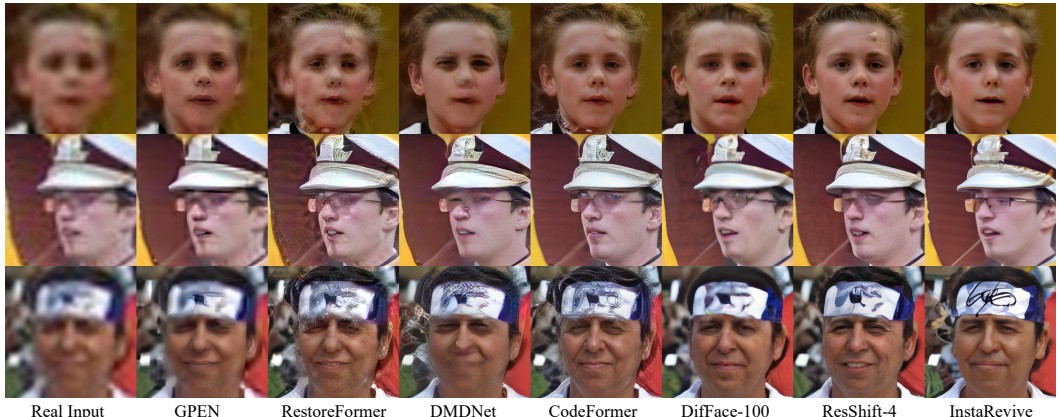

| Real Input | GPEN | RestoreFormer | DMDNet | CodeFormer | DifFace-100 | ResShift-4 | InstaRevive |

Figure 11: **More qualitative comparisons on the real-world faces.** Our method performs plausible enhancement on real-world faces, producing high-fidelity and visually satisfactory faces. Compared to other methods, InstaRevive enjoys robustness in front of challenging cases.

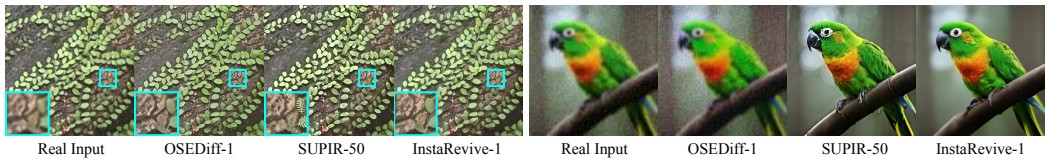

| Real Input | OSEDiff-1 | SUPIR-50 | InstaRevive-1 | Real Input | OSEDiff-1 | SUPIR-50 | InstaRevive-1 |

Figure 12: **Qualitative comparisons with recent methods.** We present visual comparisons with additional methods, demonstrating that our model consistently delivers satisfactory results.

This highlights the strength of the generator in our framework, further validating its effectiveness in addressing challenging degradations.

Table 4: **Quantitative comparison on ImageNet-Test.** Our InstaRevive demonstrates strong performance across various consistency metrics.

| Method | PSNR↑ | SSIM ↑ | LPIPS ↓ |
|---|---|---|---|
| BSRGAN (Zhang et al., 2021) | 24.42 | 0.659 | 0.259 |
| SwinIR-GAN (Liang et al., 2021) | 23.97 | 0.667 | 0.239 |
| ResShift (Yue et al., 2023) | 24.90 | 0.673 | 0.228 |
| SinSR (Wang et al., 2024) | 24.56 | 0.657 | **0.221** |
| InstaRevive (Ours) | **25.77** | **0.721** | 0.232 |

## A.3 MORE EXTENDED TASKS

Our framework is versatile and not specifically designed for any single task. To demonstrate its broader applicability, we extend our experiments to include (1) face inpainting, (2) low-light image enhancement, (3) denoising, (4) deblurring, (5) image-to-image transition, (6) deraining. We also compare part of our performance with GDP (Fei et al., 2023). Notably, our generator performs one-step inference while GDP requires 1,000 steps for sampling.

**Face inpainting**. For face inpainting, we employ the script from GPEN (Yang et al., 2021) to draw irregular polyline masks on face images as our inputs and fine-tune our BFR model for 10K iteration. During inference, we resize the mask to the latent map's shape and use it to maintain the visible area on the inputs. As shown in Figure 14, our InstaRevive successfully reconstructs the challenging cases and seamlessly completes them with coherent content.

**Low-light image enhancement**. For low-light image enhancement, we fine-tune the BSR model with 10K iterations using the LoL dataset (Wei et al., 2018). Our results, shown in Figure 14, demonstrate

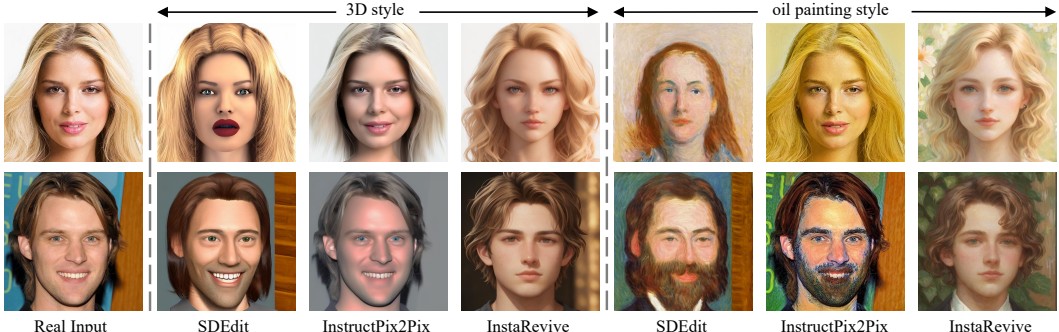

Figure 13: **More qualitative comparisons on face cartoonization.** Our InstaRevive produces high-quality results compared with other diffusion-based methods, underscoring its exceptional generalization capability for various enhancement tasks.

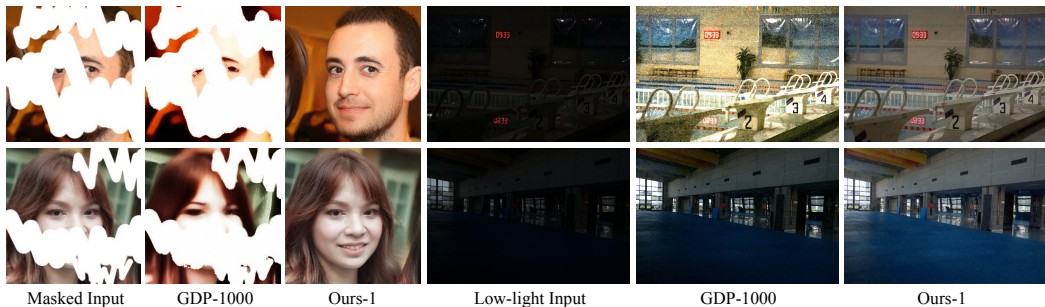

Figure 14: **Face inpainting and low-light enhancement of *InstaRevive*.** Our framework demonstrates the capability to produce satisfactory results across a range of challenging tasks.

promising outcomes with less noise compared to GDP. These findings illustrate that our method can effectively generalize to these additional challenges.

**Denoising**. To evaluate the denoising capabilities of our approach, we focus on addressing salt-and-pepper noise, a common artifact arising during imaging or data transmission. Our one-step generator is trained over 8K iterations using randomly generated salt-and-pepper noise with probabilities varying between 0.02 and 0.1. As illustrated in Row 1 of Figure 15, the proposed method effectively removes this noise while preserving the fidelity of the input content.

**Deblurring**. The trained BSR model is inherently capable of performing deblurring tasks, as it leverages the degradation model from Real-ESRGAN Wang et al. (2021c), which incorporates a variety of blur types. As demonstrated in Row 2 of Figure 15, our proposed InstaRevive effectively restores clarity to blurred images, delivering sharper details and well-defined edges.

**Image-to-image transition**. Beyond face cartoonization, a more complex challenge lies in transitioning between real-world image domains. To investigate this, we utilize the BDD100k dataset Yu et al. (2020) for training, specifically focusing on the "day-to-night" transition. Our generator is trained for 10K iterations to achieve this task. As shown in Figure 15, the proposed model generates visually realistic and temporally consistent night-time images from daytime inputs.

**Deraining.** For training, we employ the RainTrainH Yang et al. (2017), RainTrainL Yang et al. (2017), and Rain12600 Fu et al. (2017) datasets, and evaluate our framework on the Rain-100L dataset Yang et al. (2019). As illustrated in Figure 15, our approach effectively eliminates rainy regions, producing clean and visually appealing results.

A.4    PARAMETERS AND INFERENCE TIME

As noted in our limitations, our one-step generator cannot surpass larger models employing multi-step inference. However, it achieves a favorable balance between efficiency and performance. To provide a clear comparison of model parameters and inference time (evaluated on an Nvidia 3090 GPU), we

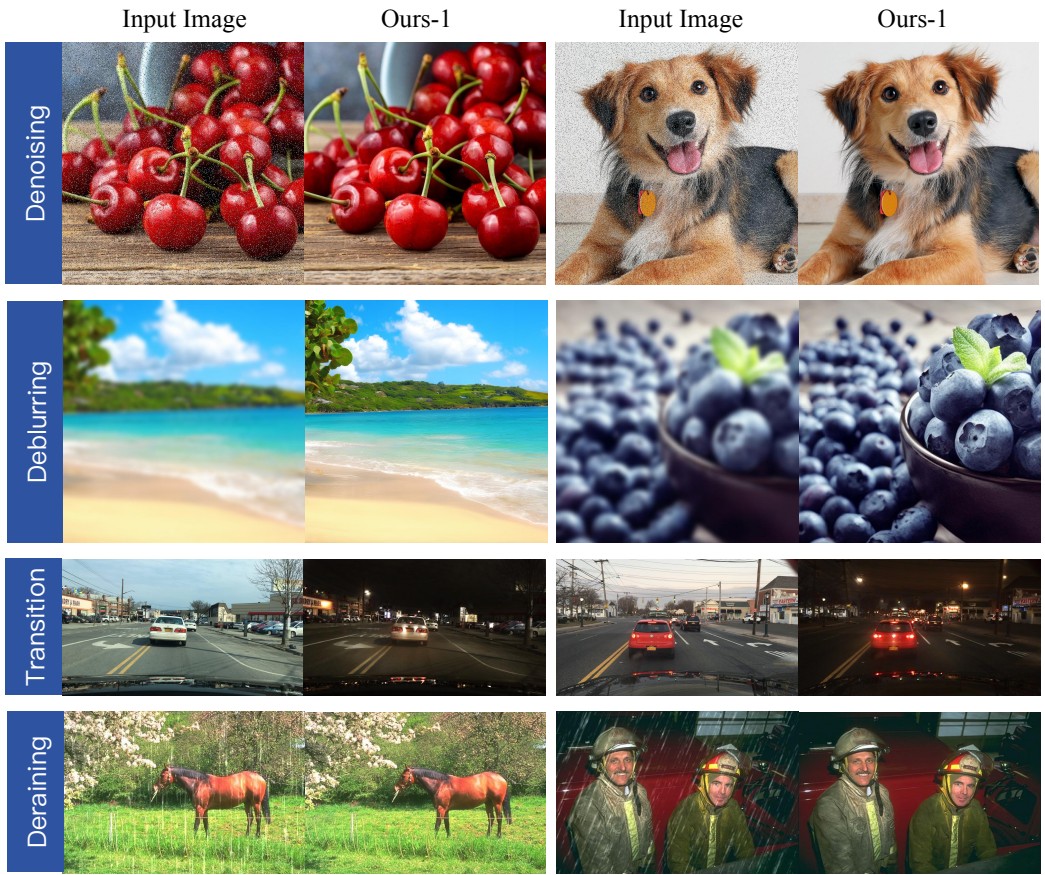

Figure 15: **More extended tasks of *InstaRevive*.** Our framework demonstrates strong performance across more tasks, including denoising, deblurring, image-to-image transitions, and deraining.

present Table 5, which includes state-of-the-art methods on RealSet65. Additionally, we fine-tune a generator based on the denoising U-Net from Stable Diffusion 2.1 and include its results for reference. As demonstrated in Table 5, our DiT-based generator achieves image quality comparable to SUPIR (Yu et al., 2024) and DiffBIR (Lin et al., 2023b), while maintaining the inference speed advantages of one-step methods. The U-Net variant also delivers competitive results, although it exhibits a minor decrease in efficiency compared to the DiT-based model. Furthermore, our method achieves comparable performance to OSEDiff (Wu et al., 2024), which incorporates additional tuning of the VAE encoder within the diffusion model.

Table 5: **Comparison on parameter and inference time.** Our generator strikes an effective balance between computational efficiency and performance.

| Method | Params. | Inference Time (s) | MANIQA↑ | MUSIQ↑ |
| --- | --- | --- | --- | --- |
| ResShift-15 (Yue et al., 2023) | 121M | 1.13 | 0.3958 | 61.33 |
| SinSR-1 (Wang et al., 2024) | 119M | 0.12 | 0.4374 | 62.64 |
| OSEDiff-1 (Wu et al., 2024) | 866M | 0.18 | 0.4573 | 65.68 |
| SUPIR-50 (Yu et al., 2024) | 3.86B | 14.88 | **0.4735** | **66.79** |
| DiffBIR-50 (Lin et al., 2023b) | 1.22B | 10.27 | 0.4612 | 66.24 |
| InstaRevive-1 (U-Net) | 865M | 0.18 | 0.4547 | 65.48 |
| InstaRevive-1 (DiT) | 611M | 0.14 | 0.4571 | 65.85 |

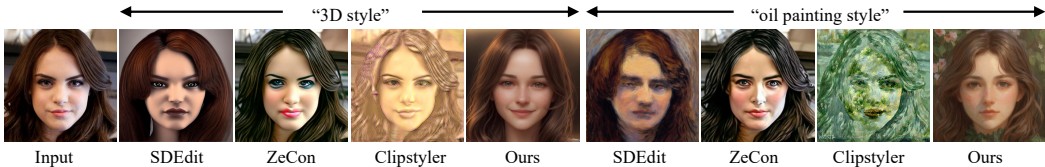

Figure 16: **Qualitative comparison with style transfer methods.** Please zoom in for the best view.

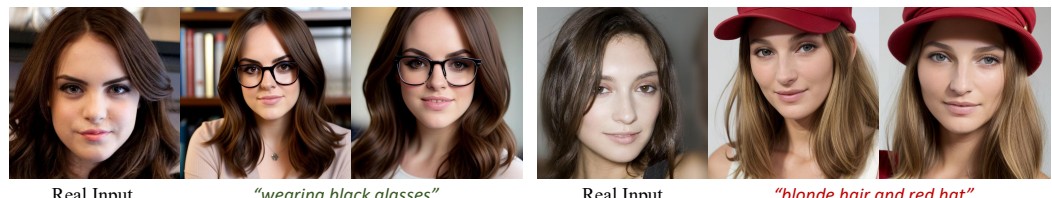

Figure 17: **Image-to-image transition with identity consistency.** The generated image maintains a similar identity to that of the input image.

## A.5 IMAGE-TO-IMAGE TRANSITION WITH IDENTITY CONSISTENCY

In tasks like image cartoonization, maintaining the source image's identity is not essential. However, in many other applications, such as face swapping and style transfer, identity consistency is crucial. Compared with style transfer methods like Clipstyler (Kwon & Ye, 2022) and ZeCon (Yang et al., 2023), our face cartoonization model falls behind in identity similarity, as shown in Figure 16. To address this, we can utilize a teacher model equipped with identity-preservation capabilities. This enables our generator to learn image-to-image transitions while retaining identity. In our experiment, we employed the IP-Adapter (Ye et al., 2023), a diffusion-based model with an image prompt adapter, as our teacher model $\epsilon_\phi$. After training with InstaRevive's framework, our generator demonstrated a proficient capacity for identity preservation, as illustrated in Figure 17.

## A.6 ABOUT DIVERSITY AND CONSISTENCY

Consistency is our primary concern in the BFR and BSR tasks. We employ a regression loss to ensure consistency and prevent unnecessary hallucinations during training. However, it is important to stress that, as discussed in works like BSRGAN (Zhang et al., 2021), these tasks are inherently ill-posed. The degradation processes in real-world scenarios are often complex and unknown, leading to irreversible damage to images. This allows various plausible results from the low-quality input. It's notable that the allowance for multiple results does not necessarily cause inconsistency as long as they align with the input. To demonstrate the diversity of the output, we provide visual results in Figure 18, showing five diverse outputs from different initial noises. These outputs include various shapes of the hat and slight differences in face identity, but all share consistency with the input.

## A.7 FAILURE CASES

InstaRevive encounters challenges when dealing with extremely challenging scenarios. As illustrated in Figure 19, our model faces difficulties with severely blurred inputs (left half) and images containing intensive objects and complex content (right half). Compared to GAN-based methods like BSRGAN (Zhang et al., 2021) which introduce many artifacts and blur, InstaRevive generates cleaner images. However, the final results may still exhibit unrealistic regions due to the challenging scenarios. Recently, some multi-step methods like SUPIR (Yu et al., 2024) attempt to handle these limitations with large-scale models that generate finer details but at the cost of significant computational resources and extended inference times. Despite these advancements, they also fall short in completely resolving these issues, sometimes leading to artificial and unrealistic regions.

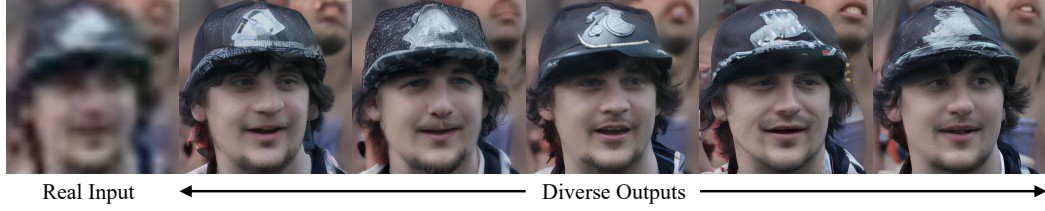

Real Input ◄——————————— Diverse Outputs ———————————►

Figure 18: **Diversity of *InstaRevive*.** Our framework can produce a range of results based on different initial noise inputs. While all outcomes are plausible, they exhibit slight variations in detail.

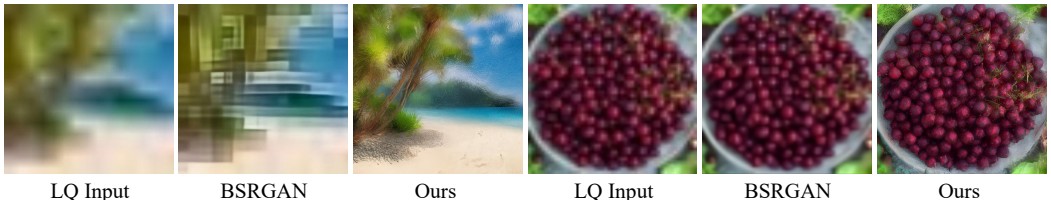

LQ Input          BSRGAN          Ours          LQ Input          BSRGAN          Ours

Figure 19: **Failure cases.** Our framework may give unsatisfactory results when confronted with severe degradation or complex content.

## A.8 THEORY OF DYNAMIC SCORE MATCHING

### A.8.1 DETAILS ABOUT SCORE MATCHING DISTILLATION

We start by expanding the KL divergence term $D_{\mathrm{KL}}(q_0||p_0)$ as:

$$D_{\mathrm{KL}}(q_0||p_0) = \mathop{\mathbb{E}}_{\boldsymbol{x}\sim q_0}\left(\log\left(\frac{q_0(\boldsymbol{x}|\boldsymbol{y})}{p_0(\boldsymbol{x}|\boldsymbol{y})}\right)\right) = \mathop{\mathbb{E}}_{\substack{\boldsymbol{x}_{\mathrm{LQ}}\sim\mathcal{X}\\\boldsymbol{x}=G_\theta(\boldsymbol{x}_{\mathrm{LQ}},\boldsymbol{y})}}\left(-\left(\log p_0(\boldsymbol{x}|\boldsymbol{y})-\log q_0(\boldsymbol{x}|\boldsymbol{y})\right)\right), \quad (8)$$

where $\mathcal{X}$ is the LQ image dataset. To find the optimal point, we calculate the gradient as follows:

$$\nabla_\theta D_{\mathrm{KL}} = \mathop{\mathbb{E}}_{\substack{\boldsymbol{x}_{\mathrm{LQ}}\sim\mathcal{X}\\\boldsymbol{x}=G_\theta(\boldsymbol{x}_{\mathrm{LQ}},\boldsymbol{y})}}\left[-\left(\nabla_{\boldsymbol{x}}\log p_0(\boldsymbol{x}|\boldsymbol{y})-\nabla_{\boldsymbol{x}}\log q_0(\boldsymbol{x}|\boldsymbol{y})\right)\frac{\partial G_\theta(\boldsymbol{x}_{\mathrm{LQ}},\boldsymbol{y})}{\partial\theta}\right], \quad (9)$$

where we denote the first two gradient terms $\nabla_{\boldsymbol{x}}\log p_0(\boldsymbol{x}|\boldsymbol{y})$ and $\nabla_{\boldsymbol{x}}\log q_0(\boldsymbol{x}|\boldsymbol{y})$ as the scores of HQ images and generated images, respectively. These scores, akin to gradients of data density, suggest the use of diffusion models for computation. Ideally, the above optimization will match the distributions of HQ images and generated results. However, the score can easily diverge when the probability term is small—specifically, $p_0(\boldsymbol{x}|\boldsymbol{y})$ vanishes when $\boldsymbol{x}$ is far away from HQ images. Another issue is that the score estimator, the diffusion model, performs best with noisy images obtained through the diffusion process. Score-SDE (Song & Ermon, 2019; Song et al., 2020) introduces a method that diffuses the original distributions with varying scales of noise indexed by $t$ and optimizes a series of KL divergences between these diffused distributions, $D_{\mathrm{KL}}(q_t||p_t)$.

Assuming that the characteristic functions of distribution $q_0$ and $p_0$ are $\phi_{q_0}(s)$ and $\phi_{p_0}(s)$. The diffusion process satisfying that $\boldsymbol{x}_t = \alpha_t\boldsymbol{x} + \sigma_t\boldsymbol{\epsilon}$, where $\boldsymbol{\epsilon}\sim\mathcal{N}(0,\boldsymbol{I})$. Considering the property of characteristic function, we obtain that:

$$\phi_{p_t}(s) = \phi_{p_0}(\alpha_t s)\phi_{\boldsymbol{\epsilon}}(\sigma_t s) = \exp(-\frac{\sigma_t^2 s^2}{2})\phi_{p_0}(\alpha_t s) \quad (10)$$

In the same way, we can get $\phi_{q_t}(s) = \exp(-\frac{\sigma_t^2 s^2}{2})\phi_{q_0}(\alpha_t s)$. Therefore, we conclude that

$$D_{\mathrm{KL}}(q_t||p_t) = 0 \Leftrightarrow q_t = p_t \Leftrightarrow \phi_{q_t} = \phi_{p_t} \Leftrightarrow \phi_{q_0} = \phi_{p_0} \Leftrightarrow q_0 = p_0 \quad (11)$$

For each $t$, $D_{\mathrm{KL}}(q_t||p_t)$ and $D_{\mathrm{KL}}(q_0||p_0)$ reach their minimum values simultaneously. So it is equivalent to minimize $D_{KL}(q_t||p_t)$. Similar to Equation equation 9, the gradient of this KL divergence is:

$$\nabla_\theta D_{\mathrm{KL}} = \mathbb{E}_{t,\boldsymbol{\epsilon},\boldsymbol{x}_{\mathrm{LQ}}} \left[ -\left( \nabla_{\boldsymbol{x}_t} \log p_t(\boldsymbol{x}_t|\boldsymbol{y}) - \nabla_{\boldsymbol{x}_t} \log q_t(\boldsymbol{x}_t|\boldsymbol{y}) \right) \frac{\partial G_\theta(\boldsymbol{x}_{\mathrm{LQ}}, \boldsymbol{y})}{\partial \theta} \right] \tag{12}$$

Additionally, we adopt the same time-dependent scalar weight $\omega(t)$ for better training dynamics. We utilize a pre-trained diffusion model $\boldsymbol{\epsilon}_\phi$ to estimate $\nabla_{\boldsymbol{x}_t} \log q_t(\boldsymbol{x}_t|\boldsymbol{y})$ and train another diffusion model $\boldsymbol{\epsilon}_\psi$ to predict $\nabla_{\boldsymbol{x}_t} \log p_t(\boldsymbol{x}_t|\boldsymbol{y})$. Therefore, the total gradient of the KL divergence becomes:

$$\nabla_\theta D_{\mathrm{KL}} = \mathbb{E}_{t,\boldsymbol{\epsilon},\boldsymbol{x}_{\mathrm{LQ}}} \left[ \omega(t)\sigma_t \left( \boldsymbol{\epsilon}_\psi(\boldsymbol{x}_t|\boldsymbol{y}) - \boldsymbol{\epsilon}_\phi(\boldsymbol{x}_t|\boldsymbol{y}) \right) \frac{\partial G_\theta(\boldsymbol{x}_{\mathrm{LQ}}, \boldsymbol{y})}{\partial \theta} \right], \tag{13}$$

### A.8.2 DETAILS ABOUT DYNAMIC NOISE CONTROL

**Predicted noise $\boldsymbol{\epsilon}_\psi$, pseudo-GT $\hat{\boldsymbol{x}}_0$ and score $\boldsymbol{s}_\psi$.** We first clarify the relationship between these three important predicted targets in diffusion models. As detailed in (Luo, 2022), we actually have $\hat{\boldsymbol{x}}_0 = (\boldsymbol{x}_t - \sigma_t\boldsymbol{\epsilon}_\psi)/\alpha_t$, and $\boldsymbol{s}_\psi = -\boldsymbol{\epsilon}_\psi/\sigma_t$. Furthermore, we have:

$$\|\boldsymbol{s}_\psi(\boldsymbol{x}_t, t, \boldsymbol{y}) - \nabla_{\boldsymbol{x}_t}\mathrm{log}p(\boldsymbol{x}_t|\boldsymbol{y})\|_2^2 = \frac{1}{\sigma_t^2}\|\boldsymbol{\epsilon}_\psi - \boldsymbol{\epsilon}_0\|_2^2 = \frac{\sqrt{\alpha_t \alpha_{t-1}^3}}{\sigma_t^4}\|\hat{\boldsymbol{x}}_0(\boldsymbol{x}_t, t) - \boldsymbol{x}_0\|_2^2 \tag{14}$$

Where $\boldsymbol{x}_0$ is GT, and $\boldsymbol{\epsilon}_0$ is the GT noise. Therefore, we conclude that the accuracy of predicted noise $\boldsymbol{\epsilon}_\psi$, pseudo-GT $\hat{\boldsymbol{x}}_0$ and score $\boldsymbol{s}_\psi$ is consistent, allowing us to determine the accuracy of the other two based on the accuracy of any one of them.

**The motivation and rationale for controlling $T_{\max}$.** As figured out in 2, the estimates of gradients for real data distribution could be inaccurate or overly smoothed when $x_t$ is far away from $x_{\mathrm{HQ}}$, so we hope to limit their mean distance. Considering the diffusion process that $\boldsymbol{x}_t = \alpha_t\boldsymbol{x} + \sigma_t\boldsymbol{\epsilon}, \boldsymbol{\epsilon} \sim \mathcal{N}(0, \boldsymbol{I})$ and using triangle inequality, we obtain that

$$\begin{aligned} \mathbb{E}_t\|\boldsymbol{x}_t - \boldsymbol{x}_{\mathrm{HQ}}\|_2 &\le \mathbb{E}_t\|\boldsymbol{x}_t - \boldsymbol{x}\|_2 + \|\boldsymbol{x}_{\mathrm{HQ}} - \boldsymbol{x}\|_2 \\ &= \mathbb{E}_t\|(\alpha_t - 1)\boldsymbol{x} + \sigma_t\boldsymbol{\epsilon}\|_2 + \|\boldsymbol{x}_{\mathrm{HQ}} - \boldsymbol{x}\|_2 \\ &\le |\alpha_t - 1|\|\boldsymbol{x}\|_2 + \sigma_t\mathbb{E}_t\|\boldsymbol{\epsilon}\|_2 + \|\boldsymbol{x}_{\mathrm{HQ}} - \boldsymbol{x}\|_2 \end{aligned} \tag{15}$$

Given that $\|\boldsymbol{x}\|_2, \mathbb{E}_t\|\boldsymbol{\epsilon}\|_2, \|\boldsymbol{x}_{\mathrm{HQ}} - \boldsymbol{x}\|_2$ are all constants, the upper bound of $\mathbb{E}_t\|\boldsymbol{x}_t - \boldsymbol{x}_{\mathrm{HQ}}\|_2$ is indeed controlled by noise schedule $\alpha_t, \sigma_t$. When $t = 0$, $\alpha_t = 1$ and $\sigma_t = 0$. As t increases, $\alpha_t$ approaches 0, while $\sigma_t$ tends to 1. Therefore, the upper bound of $\mathbb{E}_t\|\boldsymbol{x}_t - \boldsymbol{x}_{\mathrm{HQ}}\|_2$ will monotonically increase with $t$. By utilizing a limited $T_{\max}$, we can reduce the expected distance between $\boldsymbol{x}_t$ and $\boldsymbol{x}_{\mathrm{HQ}}$, thus ensuring better score estimates.

### A.9 MORE IMPLEMENTATION DETAILS

**Datasets.** For BFR, we use FFHQ (CC BY-NC-SA 4.0) (Karras et al., 2019) for training, which contains 70,000 face images in $1024 \times 1024$ resolution. For evaluation, we utilize synthetic dataset CelebA (custom, research-only, non-commercial) (Liu et al., 2015) with 3,000 HQ-LQ pairs. We also employ LFW-Test (custom, research-only) (Wang et al., 2021b) and WIDER-Test (custom, research-only) (Zhou et al., 2022) for in-the-wild evaluation. For BSR task, we use large-scale ImageNet (custom, research, non-commercial) (Deng et al., 2009) for training, and we leverage RealSR (repository link) (Cai et al., 2019) and RealSet65 (NTU S-Lab License 1.0) (Yue et al., 2023) as benchmarks.

**Training details.** The training process is efficient with our dynamic score matching strategy, which focuses on refining detailed content in LQ images. To train this framework, we utilize 4 Nvidia A800 GPUs with approximately 2.5 days. As demonstrated in Section. 3, we concurrently update the parameters of the generator $G_\theta$ and the score estimator $\boldsymbol{\epsilon}_\phi$. To achieve this, we adopt a two-stage training pipeline. Firstly, we calculate the regression loss and the KL divergence using the generated result $\boldsymbol{x}$. For the regression loss, we employ the Learned Perceptual Image Patch Similarity (LPIPS) (Zhang et al., 2018b) for better quality. The KL divergence is backpropagated as equation 5. Following this, we update the score estimator $\boldsymbol{\epsilon}_\phi$ to align the generated distribution $q_0$. We detach $\boldsymbol{x}$ and compute the diffusion loss according to equation 2. This step ensures that $\boldsymbol{\epsilon}_\phi$ accurately learns

the distribution of the generated images and provides precise scores. Note that we use two optimizers for these two models, respectively. In our framework, we mainly manipulate images in a latent space established by a trained VQGAN, which comprises an encoder $\mathcal{E}$ and a decoder $\mathcal{D}$, facilitating seamless conversion between pixel space and latent space. To compute the LPIPS, we need to convert the latent codes to pixel space with the decoder $\mathcal{D}$.

## A.10  CODE

We have included the complete source code for our method in the `./code` folder. This folder contains all the necessary files and instructions to reproduce our experiments with the InstaRevive framework.

