# OpenReview forum: "InstaRevive: One-Step Image Enhancement via Dynamic Score Matching"
_ICLR.cc/2025/Conference — ICLR 2025 Poster_

### Official Review · Reviewer_ZCNg · 2024-10-16

**Soundness:** 3
**Presentation:** 4
**Contribution:** 3
**Rating:** 6
**Confidence:** 3

**Summary:**

InstaRevive presents a novel and efficient approach to image enhancement, leveraging score-based diffusion models for impressive results. The framework introduces dynamic score matching to refine the training process and incorporates textual prompts for enhanced guidance. Experimental results demonstrate superior performance across various tasks, including blind face restoration and blind image super-resolution.

**Strengths:**

* **Clear and concise presentation:** The paper is well-structured and easy to follow, making the proposed method accessible to a wide audience.
* **Strong performance:** InstaRevive consistently outperforms baseline methods on challenging image enhancement tasks, highlighting its effectiveness.
* **Innovative approach:** The dynamic score matching technique and incorporation of textual prompts offer a unique and promising direction for image enhancement.

**Weaknesses:**

* While the paper demonstrates excellent results on face restoration and image super-resolution, it would benefit from exploring its applicability to other degradation types, such as deraining, dehazing, deblurring, and salt and pepper noise.
* The paper's focus on face cartoonization as an image-to-image translation task could be expanded to include other types of image-to-image translations, such as real image to real image translation.

**Questions:**

- How might InstaRevive need to be adapted to handle deblurring or dehazing tasks? Are there any particular challenges you foresee in extending the framework to these degradation types?

- Have you considered applying InstaRevive to style transfer tasks between real images? What modifications, if any, would be necessary to adapt the framework for tasks like day-to-night image conversion or season transfer?

---

> ### Author Response · Authors · 2024-11-23
>
> We thank Reviewer ZCNg for the valuable feedback. We address your concerns and questions below.
> >### **Q1: About more extend tasks**
>
> **[Reply]:** Thank you for your valuable suggestion. We have conducted further experiments to evaluate the applicability of our method to additional tasks, including various degradation types and real image-to-image transitions. In Section A.3 of the appendix (Page 19), we have added four new tasks: (1) denoising (salt-and-pepper noise), (2) deblurring, (3) image-to-image transition (day-to-night), and (4) deraining. Due to time constraints, we provide qualitative results for these tasks in Figure 15 (Page 20). These results demonstrate the versatility and adaptability of our framework to a broader range of applications.
>
> >### **Q2: About modifications for other tasks**
>
> **[Reply]:** As discussed earlier, our framework can produce plausible results for a variety of tasks without requiring modifications. However, we did encounter certain challenges when extending to these tasks.
> First, our method depends on a substantial number of paired images for generator tuning, and the lack of suitable data can limit its performance in tasks like deraining and style transfer. Second, maintaining consistency with the source image, such as preserving spatial layouts and objects, poses additional challenges, particularly in style transfer tasks. To address these issues, we propose the following potential solutions:
>
> - **Enhanced Image Conditioning**: Utilizing more advanced image conditioning techniques, such as ControlNet and IP-Adapter, can significantly improve consistency by strengthening the image condition injection. We explored integrating IP-Adapter in Section A.5 (Page 21), which helped preserve facial identity in source images.
>
> - **Self-Supervised Learning**: By relying solely on the dynamic score matching loss in our framework, self-supervised learning without paired data is achievable. However, we observed that this approach can result in unstable loss curves and convergence issues. Further research into stabilizing and refining this technique will be a focus of our future work.

---

> > ### Comment · Reviewer_ZCNg · 2024-11-26
> >
> > Thank you for your thoughtful response and for sharing the new results. I noticed in Fig. 13 that the proposed method appears to struggle with preserving identity compared to the other methods. Could you explain the potential reason for this?

---

> ### Author Response · Authors · 2024-11-26
>
> >### **About the identity consistency**
>
> Thank you for your observation. Yes, we acknowledge that the original implementation of face cartoonization struggles to preserve facial identity. We attribute this to several factors:
> - **Dataset Limitations**: The main reason lies in the dataset we employed. To produce the stylized face dataset from the FFHQ dataset, we utilized a common diffusion-based image-to-image model. While this model effectively transfers styles, it fails to retain the source identity, leading to a generator trained on this dataset lacking the capability to preserve identity. We believe that using a more specialized model, such as InstantID [a], to construct the dataset could significantly improve identity consistency.
> - **Simple Image Conditioning**: Our current approach involves concatenating the input image with the latent features at the generator’s initial stage. While this straightforward approach is effective for tasks like image restoration and super-resolution, it can lead to the loss of high-level spatial features, including those critical for preserving identity. Recent works, such as IP-Adapter and InstantID, have demonstrated that maintaining facial identity in image-to-image tasks often requires incorporating a specialized network with cross-attention layers. These layers infuse identity features extracted by a face encoder into the latent features at every block of the denoising model. The absence of such advanced architectural designs in our implementation limits identity preservation.
> - **Loss Function Design**: Our generator’s loss design, originally tailored for the blind face restoration (BFR) task, employs regression loss and dynamic score matching loss. With this setup, we achieve highly competitive identity similarity (IDS) scores, as reported in Table 1 (Page 7). However, tasks like face swapping [b, c], which especially emphasize identity consistency, typically use an identity loss to measure discrepancies in identity features between the input and output. As our framework is end-to-end, incorporating identity loss could enhance identity consistency.
>
> Our primary goal in exploring this extended task was to evaluate the generalization capability of our framework across diverse tasks. While face cartoonization prioritizes style transformation over identity preservation, we believe improvements can be achieved by:
> - Adopting identity-consistent datasets.
> - Introducing more advanced architectures.
> - Designing task-specific loss functions.
>
> As demonstrated in Section A.5 and Figure 17 (Page 21), integrating IP-Adapter into our framework has proven effective in preserving identity during image-to-image transitions. We are optimistic that further experiments along these lines will yield better identity consistency in face cartoonization.
>
> [a] "Instantid: Zero-shot identity-preserving generation in seconds." arXiv, 2024.
>
> [b] "Simswap: An efficient framework for high fidelity face swapping." ACM MM, 2020.
>
> [c] "HifiFace: 3D shape and semantic prior guided high fidelity face swapping." IJCAI, 2021

---

> > ### Comment · Reviewer_ZCNg · 2024-11-27
> >
> > Thank you for your detailed answers. I will keep my score unchanged.

---

> > > ### Author Response · Authors · 2024-11-28
> > >
> > > We sincerely appreciate your positive feedback and thoughtful recognition of our work. Your valuable suggestions will greatly assist us in improving the quality of our paper.

---

### Official Review · Reviewer_LCnj · 2024-11-02

**Soundness:** 3
**Presentation:** 4
**Contribution:** 2
**Rating:** 6
**Confidence:** 4

**Summary:**

This method improves image enhancement by utilizing a distillation approach, which significantly reduces time and computational resources. It introduces a dynamic score matching framework that effectively guides models to generate realistic (using KL divergence) and well-resolved (using regression) images from low-quality inputs. Additionally, the method is adaptable to various applications beyond BFR and BSR, including style transfer, low-light enhancement, inpainting, and more.

**Strengths:**

1. "Introducing one-step restoration using diffusion models" is a compelling topic for researchers in the field of image processing.

2. The experimental results strongly support the authors' claims and demonstrate impressive restoration outcomes for real degraded images.

3. Dynamic score matching can be valuable for various distillation studies to address potential inaccuracies at large timesteps.

**Weaknesses:**

1. While the method demonstrates impressive results, my main concern is that recent diffusion-based image restoration methods primarily focus on a zero-shot approach. My concerns are as follows:

* The proposed method initially relies on a restorer, such as SwinIR-GAN, to enhance low-quality images. This approach implies that degradation types vary, requiring the model to use different restoration modules depending on the specific degradation family.
* Additionally, the model is trained in a supervised manner using paired low-quality and high-quality datasets.


2. More comparisons with the original restorer are needed.

* From my perspective, the initial restorer recovers most of the information, while the distilled model further refines it to appear more realistic. If this is the case, it is crucial to compare the method with the initial restorer, especially using traditional metrics like PSNR, SSIM, and LPIPS. However, most experimental results do not provide this comparison.

**Questions:**

1. Why is it necessary to use a restorer-fused method rather than zero-shot diffusion-based restoration methods, such as DPS, PSLD, or GDP?

2. If possible, please provide additional comparisons with the restorer using traditional metrics.

---

> ### Author Response · Authors · 2024-11-23
>
> We sincerely thank Reviewer LCnj for the valuable insights and constructive feedback. Below, we address your concerns and questions in detail.
> >### **About comparisons with the zero-shot methods**
>
> **[Reply]:** Thank you for your insight. Zero-shot methods such as DDRM, DDNM, DPS, GDP, and PSLD indeed provide valuable approaches by leveraging pre-trained diffusion models for various tasks, including super-resolution, deblurring, and inpainting, using a single model without requiring fine-tuning.
> However, these methods also face inherent limitations that affect their practical applicability:
> - **Performance**: Due to the absence of model fine-tuning, their performance is generally lower compared to tuned methods.
> - **Inference Speed**: They rely heavily on diffusion sampling, resulting in significantly longer inference times (e.g., GDP requires 1000 steps, DPS over 250 steps, and PSLD around 200 steps).
>
> In contrast, approaches that involve fine-tuning diffusion models—such as ours, DiffBIR, ResShift, and SinSR—achieve substantially better performance and faster inference, although they require task-specific restorers. We believe this trade-off highlights the strengths of our method in delivering efficient and high-quality results for real-world applications. As shown in Table 2 and Figure 14 (Page 19), the zero-shot GDP struggles to deliver satisfactory performance in complex scenarios, whereas our fine-tuned model consistently produces robust and reliable results.
>
> >### **About the initial restorer**
>
> **[Reply]:** Thanks for your suggestions. We implemented the initial restorer to enable the recovery of coarse information for both the text captioner and the generator. However, as shown in Figures 5 and 10, as well as Table 2, this restorer (SwinIR-GAN) provides only limited enhancement, resulting in noticeable blurring and suboptimal quantitative metrics. To further support this observation, we evaluated the performance of the initial restorer (SwinIR-GAN) on the ImageNet-Test dataset, using metrics such as PSNR, SSIM, LPIPS, CLIPIQA, and MUSIQ, as reported in Table 4 of the appendix (Page 18). The results highlight the generator’s ability to significantly improve image quality and consistency with the ground truth, as reflected by higher PSNR, SSIM, and CLIPIQA scores. These findings underscore the critical role of the distilled generator in producing more realistic and consistent results:
> |Method | PSNR$\uparrow$ |  SSIM $\uparrow$ | LPIPS $\downarrow$| CLIPIQA $\uparrow$| MUSIQ $\uparrow$|
> | :-----        |    :------:   |    :------: | :------: |:------: | :------: |
> |SwinIR-GAN| 23.97 | 0.667| 0.239 | 0.564| 53.790|
> |InstaRevive (Ours) | **25.77**| **0.721**|  **0.232**| **0.620** | **54.763**|

---

> > ### Comment · Reviewer_LCnj · 2024-11-25
> >
> > I sincerely appreciate the authors for taking the time to provide their detailed and thoughtful responses. All of my concerns have been thoroughly addressed, and I will be increasing my score accordingly.

---

> > > ### Author Response · Authors · 2024-11-25
> > >
> > > Thank you for your positive feedback on our work. It has been our pleasure to address your concerns. Your valuable insights will greatly contribute to improving the quality of our paper.

---

### Official Review · Reviewer_mTLA · 2024-11-03

**Soundness:** 3
**Presentation:** 3
**Contribution:** 3
**Rating:** 6
**Confidence:** 4

**Summary:**

This paper propose a one-step diffusion framework called InstaRevive which employs score-based diffusion distillation with dynamic noise control and dynamic loss control. Dynamic noise control is to control the max noise level during training according to the difference of LQ and GT, and the dynamic loss control is to adjust the loss weights of fidelity and distribution. Then, this paper verifies the effectiveness of the framework on several enhancement tasks and generally performs well.

**Strengths:**

1. This paper proposes a dynamic control method to constrain the optimization process according to the image distribution, which effectively improves the performance of the present method.
2. The writing of the article is good, making the whole article clear and easy to understand
3. This paper verifies the effectiveness of the method on some different enhancement tasks.

**Weaknesses:**

1. The paper does not mention some important pre-work (e.g., img2img-turbo and OSEDiff), which may exaggerate some contributions of this work. In fact, the methods (Reduce the steps by distillation) have been used before, and the proposed innovation core of this paper should be the dymic control.
2. Due to the lack of reference to some previous work, in the comparison of other methods in the article are not new enough. For example, the visual effects and quantitative results of OSEDiff, SUPIR should be included.
3. The ablation of dynamic control in the experiment shown in Figure 7 is not detailed enough. Because the main contribution is dynamic control, I suggest to draw two lines for the dynamic control of noise and the dynamic control of loss, instead of coupling them together, so that we can see which method is actually taking effect.

The article has more advantages than disadvantages on the whole, but I think the above weaknesses still need to be improved.

**Questions:**

Please see weaknesses.

---

> ### Author Response · Authors · 2024-11-23
>
> We thank Reviewer mTLA for the valuable feedback. We address your concerns and questions below.
> >### **Discussion and comparisons with recent methods**
>
> **[Reply]:** Thank you for your suggestion and for highlighting relevant works. We acknowledge that recent works like OSEDiff and img2img-turbo share similarities with our framework in employing diffusion distillation. However, we would like to emphasize that our work specifically addresses challenges encountered during score distillation, such as inaccuracies and over-smoothing, as demonstrated in Figure 2. To provide a more comprehensive comparison, we have included both qualitative and quantitative results in Figure 12 (Page 18) and Table 5 (Page 20), alongside further discussions in Sections A.1 (Page 16) and A.4 (Page 19). Furthermore, we have also provided comparisons with SUPIR in the same figure and table for additional context:
>
>  |  Method | Params. | Inference Time (s)	| MANIQA $\uparrow$| MUSIQ$\uparrow$|
>  | :-----        |    :------:   |    :------: | :------:   |    :------: |
>  |SinSR-1 | 119M | 0.12 | 0.4374 | 62.64 |
>  |OSEDiff-1|  866M | 0.18 | 0.4573|	65.68|
>  |SUPIR-50 | 3.86B | 14.88 | **0.4735** | **66.79** |
>  |DiffBIR-50 |  1.22B | 10.27| 0.4612| 66.24|
>  | InstaRevive-1 | 611M |	0.14	|0.4571|	65.85|
>
> >### **About more details in Figure 7**
>
> **[Reply]:** Thanks for your valuable advice. We agree that presenting two additional curves (without noise control and without loss control) in Figure 7, as you suggested, makes the analysis more detailed and clear. We have reviewed our training logs and updated Figure 7 accordingly. As shown in the revised figure, both dynamic control approaches significantly enhance loss convergence. We appreciate your suggestion, as it helps to make our paper more comprehensive.

---

> > ### Comment · Reviewer_mTLA · 2024-11-25
> >
> > I appreciate the authors' detailed feedback. I believe it is necessary to include these results and comparisons. We have also reached a consensus that the authors address certain challenges encountered in score distillation rather than proposing an entirely new score distillation paradigm. Therefore, I will maintain my score without increasing it. Overall, I believe this paper deserves to be presented at ICLR.

---

> > > ### Author Response · Authors · 2024-11-26
> > >
> > > Thank you for your positive feedback and recognition of our work. We truly appreciate your thoughtful suggestions, which will help further enhance the quality of our paper.

---

### Author Response · Authors · 2024-11-23

We sincerely thank all reviewers for their constructive feedback and valuable suggestions. We are encouraged by the positive reception of our work, with all reviewers finding it well-written. Reviewer ZCNg described our approach as “innovative”, Reviewer mTLA acknowledged that it “effectively improves the performance”, and Reviewer LCnj highlighted its value for “various distillation studies”. We have carefully addressed the raised concerns and clarified potential confusions by incorporating corresponding modifications into our paper (with revisions highlighted in blue).

---

### Comment · Area_Chair_2M32 · 2024-11-25
**Please check the authors' responses**

Dear reviewers,

Could you please check the authors' responses, and post your message for discussion or changed scores?

best,

AC

---

### Meta-Review · Area_Chair_2M32 · 2024-12-21

**Metareview:**

This paper proposed an image enhancement method based on diffusion distillation using dynamic control, including the dynamic noise control and dynamic loss control, with additional text prompt. The proposed approach was applied to blind face restoration,  super-resolution, and more extended image enhancement tasks. The quantitative comparisons demonstrate the effectiveness of this work. The major contribution of this work is on the proposed dynamic control in the diffusion distillation. Though the reviewers raised questions on some more experimental justifications (e.g., extension to other enhancement tasks, style transfer, comparisons with zero-shot methods, etc.), the final recommendation of reviewers tend to be positive on the contribution, and the scores are all 6. The paper can be accepted, but the authors should carefully incorporate the responses to reviewers' comments to the final version, and discuss on the limitation of face identity consistency problems.

**Additional Comments On Reviewer Discussion:**

Reviewer mTLA  raised questions on missing related works, ask for clarifying the novelty accurately, including results of  OSEDiff, SUPIR, and more details on ablation study. These questions are important and the authors should include these revisions in the final version. Reviewer LCnj concerns on  the model depending on the degradation type, comparison with the baseline restorer. These concerns have been addressed in the rebuttal.  Reviewer ZCNg questioned on the extensions to more degradation types, and other image-to-image translations. After rebuttal, the reviewer kept the score of 6.  As a summary, all the raised concerns seem to be addressed in the rebuttal, and the reviewers tend to be positive in their final decisions.

---

### Decision · Program_Chairs · 2025-01-22

Accept (Poster)